# SeKron: A Decomposition Method Supporting Many Factorization Structures

## Abstract

While convolutional neural networks (CNNs) have become the de facto standard for most image processing and computer vision applications, their deployment on edge devices remains challenging. Tensor decomposition methods provide a means of compressing CNNs to meet the wide range of device constraints by imposing certain factorization structures on their convolution tensors. However, being limited to the small set of factorization structures presented by state-of-the-art decomposition approaches can lead to sub-optimal performance. We propose SeKron, a novel tensor decomposition method that offers a wide variety of factorization structures, using sequences of Kronecker products. The flexibility of SeKron leads to many compression rates and also allows it to cover commonly used factorizations such as Tensor-Train (TT), Tensor-Ring (TR), Canonical Polyadic (CP) and Tucker. Crucially, we derive an efficient convolution projection algorithm shared by all SeKron structures, leading to seamless compression of CNN models. We validate our approach for model compression on both high-level and low-level computer vision tasks and find that it outperforms state-of-the-art decomposition methods.

## 1 Introduction

Deep learning models have introduced new state-of-the-art solutions to both high-level computer vision problems (He et al. 2016; Ren et al. 2015), and low-level image processing tasks (Wang et al. 2018b; Schuler et al. 2015; Kokkinos & Lefkimmiatis 2018) through convolutional neural networks (CNNs). Such models are obtained at the expense of millions of training parameters that come along deep CNNs making them computationally intensive. As a result, many of these models are of limited use as they are challenging to deploy on resource-constrained edge devices. Compared with neural networks for high-level computer vision tasks (e.g., ResNet-50 (He et al. 2016)), models for low-level imaging problems such as single image super-resolution have much a higher computational complexity due to the larger feature map sizes. Moreover, they are typically infeasible to run on cloud computing servers. Thus, their deployment on edge devices is even more critical.

In recent years an increasing trend has begun in reducing the size of state-of-the-art CNN backbones through efficient architecture designs such as Xception (Chollet 2017), MobileNet (Howard et al. 2019), ShuffleNet (Zhang et al. 2018c), and EfficientNet (Tan & Le 2019), to name a few. On the other hand, there have been studies demonstrating significant redundancy in the parameters of large CNN models, implying that in theory the number of model parameters can be reduced while maintaining performance (Denil et al. 2013). These studies provide the basis for the development of many model compression techniques such as pruning (He et al. 2020), quantization (Hubara et al. 2017), knowledge distillation (Hinton et al. 2015), and tensor decomposition (Phan et al. 2020).

Tensor decomposition methods such as Tucker (Kim et al. 2016), Canonical Polyadic (CP) (Lebedev et al. 2015), Tensor-Train (TT) (Novikov et al. 2015) and Tensor-Ring (TR) (Wang et al. 2018a) rely on finding low-rank approximations of tensors under some imposed factorization structure as illustrated in Figure 1a. In practice, some structures are more suitable than others when decomposing tensors. Choosing from a limited set of factorization structures can lead to sub-optimal compressions as well as lengthy runtimes depending on the hardware. This limitation can be alleviated by reshaping tensors prior to their compression to improve performance as shown in (Garipov et al. 2016). However, this approach requires time-consuming development of customized convolution algorithms.

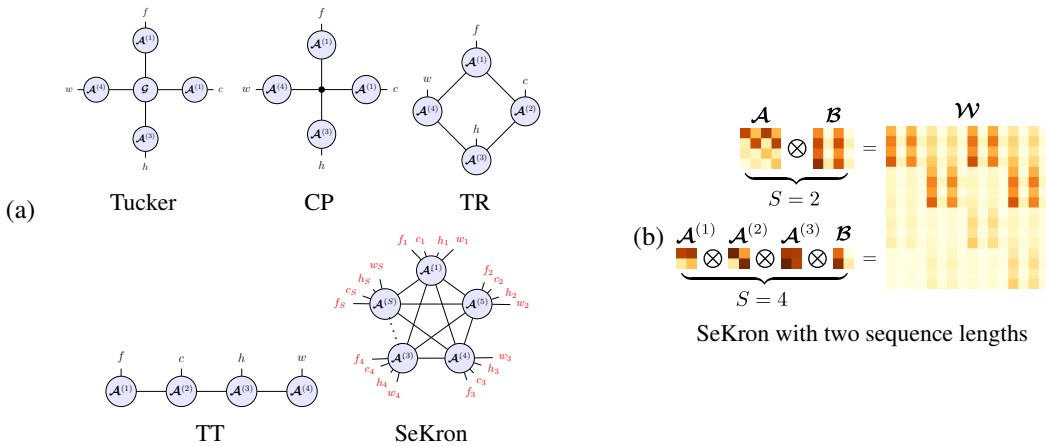

Figure 1: (a): Tensor network diagrams of various decomposition methods for a 4D convolution tensor $\mathcal{W} \in \mathbb{R}^{F \times C \times K_h \times K_w}$. Unlike all other decomposition methods where $f, c, h, w$ index over **fixed** dimensions (i.e., dimensions of $\mathcal{W}$), SeKron is flexible in its factor dimensions, with $f_k, c_k, h_k, w_k, \forall k \in \{1, ..., S\}$ indexing over **variable** dimension choices, as well as its sequence length $S$. Thus, it allows for a wide range of factorization structures to be achieved. (b): Example of a $16 \times 16$ tensor $\mathcal{W}$ that can be more efficiently represented using a sequence of four Kronecker factors (requiring **16 parameters**) in contrast to using a sequence length of two (requiring **32 parameters**).

We propose SeKron, a novel tensor decomposition method offering a wide range of factorization structures that share the *same* efficient convolution algorithm. Our method is inspired by approaches based on the Kronecker Product Decomposition (Thakker et al. 2019; Hameed et al. 2022). Unlike other decomposition methods, Kronecker Product Decomposition generalizes the product of smaller factors from vectors and matrices to a range of tensor shapes, thereby exploiting local redundancy between arbitrary slices of multi-dimensional weight tensors. SeKron represents tensors using *sequences* of Kronecker products to compress convolution tensors in CNNs. Using sequences of Kronecker products leads to a wide range of factorization structures including commonly used ones such as Tensor-Train (TT), Tensor-Ring (TR), Canonical Polyadic (CP) and Tucker.

Sequences of Kronecker products also have the potential to exploit local redundancies using far fewer parameters as illustrated in the example in Figure 1b. By performing the convolution operation using each of the Kronecker factors independently, the number of parameters, computational intensity, and runtime are reduced, simultaneously. Leveraging the flexibility SeKron, we find efficient factorization structures that outperform existing decomposition methods on various image classification and low-level image processing super-resolution tasks. In summary, our contributions are:

- Introducing SeKron, a novel tensor decomposition method based on sequences of Kronecker products that allows for a wide range of factorization structures.

- Providing a solution to the problem of finding the summation of sequences of Kronecker products between factor tensors that well approximates the original tensor.

- Deriving a single convolution algorithm shared by all factorization structures achievable by SeKron, utilized as compressed convolutional layers in CNNs.

- Improving the state-of-the-art of low-rank model compression on image classification (high-level vision) benchmarks such as ImageNet and CIFAR-10, as well as super-resolution (low-level vision) benchmarks such as Set4, Set14, B100 and Urban100.

## 2 RELATED WORK ON DNN MODEL COMPRESSION

**Sparsification.** Different components of DNNs, such as weights (Han et al. 2015b;a), convolutional filters (He et al. 2018; Luo et al. 2017) and feature maps (He et al. 2017; Zhuang et al. 2018) can be sparse. The sparsity can be enforced using sparsity-aware regularization (Liu et al. 2015; Zhou et al. 2016) or pruning techniques (Luo et al. 2017; Han et al. 2015b). Many pruning methods (Luo

et al. 2017; Zhang et al. 2018b) aim for a high compression ratio and accuracy regardless of the structure of the sparsity. Thus, they often suffer from imbalanced workload caused by irregular memory access. Hence, several works aim at zeroing out structured groups of DNN components through more hardware friendly approaches (Wen et al. 2016).

**Quantization.** The computation and memory complexity of DNNs can be reduced by quantizing model parameters into lower bit-widths; wherein the majority of research works use fixed-bit quantization. For instance, the methods proposed in (Gysel et al. 2018; Louizos et al. 2018) use fixed 4 or 8-bit quantization. Model parameters have been quantized even further into ternary (Li et al. 2016; Zhu et al. 2016) and binary (Courbariaux et al. 2015; Rastegari et al. 2016; Courbariaux et al. 2016), representations. These methods often achieve low performance even with unquantized activations (Li et al. 2016). Mixed-precision approaches, however, achieve more competitive performance as shown in (Uhlich et al. 2019) where the bit-width for each layer is determined in an adaptive manner. Also, choosing a uniform (Jacob et al. 2018) or nonuniform (Han et al. 2015a; Tang et al. 2017; Zhang et al. 2018a) quantization interval has important effects on the compression rate and the acceleration.

**Tensor Decomposition.** Tensor decomposition approaches are based on factorizing weight tensors into smaller tensors to reduce model sizes (Yin et al. 2021). Singular value decomposition (SVD) applied on matrices as a 2-dimensional instance of tensor decomposition is used as one of the pioneering approaches to perform model compression (Jaderberg et al. 2014). Other classical high-dimensional tensor decomposition methods, such as Tucker (Tucker 1963) and CP decomposition (Harshman et al. 1970), are also adopted to perform model compression. However, using these methods often leads to significant accuracy drops (Kim et al. 2015; Lebedev et al. 2015; Phan et al. 2020). The idea of reshaping weights of fully-connected layers into high-dimensional tensors and representing them in TT format (Oseledets 2011) was extended to CNNs in (Garipov et al. 2016). For multidimensional tensors, TR decomposition (Wang et al. 2018a) has become a more popular option than TT (Wang et al. 2017). Subsequent filter basis decomposition works polished these approaches using a shared filter basis. They have been proposed for low-level computer vision tasks such as single image super-resolution in (Li et al. 2019). Kronecker factorization is another approach to replace the weight tensors within fully-connected and convolution layers (Zhou et al. 2015). The rank-1 Kronecker product representation limitation of this approach is alleviated in (Hameed et al. 2022). The compression rate in (Hameed et al. 2022) is determined by both the rank and factor dimensions. For a fixed rank, the maximum compression is achieved by selecting dimensions for each factor that are closest to the square root of the original tensors' dimensions. This leads to representations with more parameters than those achieved using sequences of Kronecker products as shown in Fig. 1b.

There has been extensive research on tensor decomposition through characterizing global correlation of tensors (Zheng et al. 2021), extending CP to non-Gaussian data (Hong et al. 2020), employing augmented decomposition loss functions (Afshar et al. 2021), etc. for different applications. Our main focus in this paper is on the ones used for NN compression.

**Other Methods** NNs can also be compressed using Knowledge Distillation (KD) where a large pre-trained network known as teacher is used to train a smaller student network (Mirzadeh et al. 2020; Heo et al. 2019). Sharing weights in a more structured manner can be another model compression approach as FSNet (Yang et al. 2020) which shares filter weights across spatial locations or ShaResNet (Boulch 2018) which reuses convolutional mappings within the same scale level. Designing lightweight CNNs (Sandler et al. 2018; Iandola et al. 2016; Chollet 2017; Howard et al. 2019; Zhang et al. 2018c; Tan & Le 2019) is another direction orthogonal to the aforementioned approaches.

# 3 METHOD

In this section, we introduce SeKron and how it can be used to compress tensors in deep learning models. We start by providing background on the Kronecker Product Decomposition in Section 3.1. Then, we introduce our decomposition method in 3.2. In Section 3.3, we provide an algorithm for computing the convolution operation using each of the factors directly (avoiding reconstruction) at runtime. Finally, we discuss the computational complexity of the proposed method in Section 3.4.

### 3.1 PRELIMINARIES

Convolutional layers prevalent in CNNs transform an input tensor $\mathcal{X} \in \mathbb{R}^{C \times K_h \times K_w}$ using a weight tensor $\mathcal{W} \in \mathbb{R}^{F \times C \times K_h \times K_w}$ via a multi-linear map given by

$$\mathcal{Y}_{f,x,y} = \sum_{i=1}^{K_h} \sum_{j=1}^{K_w} \sum_{c=1}^{C} \mathcal{W}_{f,c,i,j} \mathcal{X}_{c,i+x,j+y}, \tag{1}$$

where $C$ and $F$ denote the number of input channels and output channels, respectively, and $K_h \times K_w$ denotes the spatial size of the weight (filter).

Tensor decomposition seeks an approximation to replace $\mathcal{W}$, typically through finding lower-rank tensors using SVD. One such approximation comes from the fact that any tensor $\mathcal{W} \in \mathbb{R}^{w_1 \times \cdots \times w_N}$ can be written as a sum of Kronecker products (i.e., $\mathcal{W} = \sum_{r=1}^{R} \mathcal{A}_r \otimes \mathcal{B}_r$, where $= \mathcal{A}_r \in \mathbb{R}^{a_1 \times \cdots \times a_N}$, $\mathcal{B}_r \in \mathbb{R}^{b_1 \times \cdots \times b_N}$ and $a_j b_j = w_j$ for $j \in \{1, \cdots, N\}$). (Hameed et al. 2022). Thus, a lower-rank approximation can be obtained by solving

$$\min_{\{\mathcal{A}_r\}, \{\mathcal{B}_r\}} \left\| \mathcal{W} - \sum_{r=1}^{\widehat{R}} \mathcal{A}_r \otimes \mathcal{B}_r \right\|_{\mathrm{F}}^2, \tag{2}$$

for $\widehat{R}$ sums of Kronecker products ($\widehat{R} \le R$) using the SVD of a particular reshaping (unfolding) of $\mathcal{W}$, where $\|\cdot\|_{\mathrm{F}}$ denotes the Frobenius norm.

### 3.2 SEKRON TENSOR DECOMPOSITION

The Kronecker decomposition in equation 2 can be extended to finding an approximating *sequence* of Kronecker factors $\mathcal{A}^{(k)} \in \mathbb{R}^{R_1 \times \cdots \times R_k \times a_1^{(k)} \times \cdots \times a_N^{(k)}}$ as follows:

$$\min_{\{\mathcal{A}^{(k)}\}_{k=1}^S} \left\| \mathcal{W} - \sum_{r_1=1}^{R_1} \left( \mathcal{A}_{r_1}^{(1)} \otimes \sum_{r_2=1}^{R_2} \left( \mathcal{A}_{r_1 r_2}^{(2)} \otimes \cdots \otimes \sum_{r_{S-1}=1}^{R_{S-1}} \mathcal{A}_{r_1 \cdots r_{S-1}}^{(S-1)} \otimes \mathcal{A}_{r_1 \cdots r_{S-1}}^{(S)} \right) \right) \right\|_{\mathrm{F}}^2. \tag{3}$$

Although this is a non-convex objective, a quasi-optimal solution based on recursive application of SVD is given in Theorem 1. Note that alternative expansion directions to 3 are viable (See Appendix B).

**Theorem 1** (Tensor Decomposition using a Sequence of Kronecker Products). *Any tensor $\mathcal{W} \in \mathbb{R}^{w_1 \times \cdots \times w_N}$ can be represented by a sequence of Kronecker products between $S \in \mathbb{N}$ factors:*

$$\mathcal{W} = \sum_{r_1=1}^{R_1} \mathcal{A}_{r_1}^{(1)} \otimes \sum_{r_2=1}^{R_2} \mathcal{A}_{r_1 r_2}^{(2)} \otimes \cdots \otimes \sum_{r_{S-1}=1}^{R_{S-1}} \mathcal{A}_{r_1 \cdots r_{S-1}}^{(S-1)} \otimes \mathcal{A}_{r_1 \cdots r_{S-1}}^{(S)}, \tag{4}$$

*where $R_i \in \mathbb{N}$ and $\mathcal{A}^{(k)} \in \mathbb{R}^{R_1 \times \cdots \times R_k \times a_1^{(k)} \times \cdots \times a_N^{(k)}}$.*

*Proof.* See Appendix C $\qquad\qquad\qquad\qquad\qquad\qquad\qquad\qquad\qquad\qquad\qquad\qquad\qquad\qquad$ □

Our approach to solving equation 3 involves finding two approximating Kronecker factors that minimize the reconstruction error with respect to the original tensor, then recursively applying this procedure on the latter factor found. More precisely, we define intermediate tensors

$$\mathcal{B}_{r_1 \cdots r_k}^{(k)} \triangleq \sum_{r_{k+1}=1}^{R_{k+1}} \mathcal{A}_{r_1 \cdots r_{k+1}}^{(k+1)} \otimes \sum_{r_{k+2}=1}^{R_{k+2}} \mathcal{A}_{r_1 \cdots r_{k+2}}^{(k+2)} \otimes \cdots \otimes \sum_{r_{S-1}=1}^{R_{S-1}} \mathcal{A}_{r_1 \cdots r_{S-1}}^{(S-1)} \otimes \mathcal{A}_{r_1 \cdots r_{S-1}}^{(S)}, \tag{5}$$

allowing us to re-write the reconstruction error in equation 3, for the $k^{th}$ iteration, as

$$\min_{\substack{\{\mathcal{A}_{r_1 \cdots r_k}^{(k)}, \mathcal{B}_{r_1 \cdots r_k}^{(k)}\} \\ r_j=1, \cdots R_j, \, j=1, \ldots, k}} \left\| \mathcal{W}_{r_1 \cdots r_{k-1}}^{(k)} - \sum_{r_k=1}^{R_k} \mathcal{A}_{r_1 \cdots r_k}^{(k)} \otimes \mathcal{B}_{r_1 \cdots r_k}^{(k)} \right\|_{\mathrm{F}}^2. \tag{6}$$

---

**Algorithm 1:** SeKron Tensor Decomposition

---

**Input:** Input tensor $\mathcal{W} \in \mathbb{R}^{w_1 \times \cdots \times w_N}$  Kronecker factor shapes $\{d^{(i)}\}_{i=1}^S$
**Output:** Kronecker factors $\{\mathcal{A}\}_{i=1}^S$
**for** $i \leftarrow 1, 2, \ldots, S-1$ **do**

    $\mathbf{d}^{(a)} \leftarrow \mathbf{d}^{(i)}$

    $\mathbf{d}^{(b)} \leftarrow \prod_{k=i+1}^S \mathbf{d}^{(k)}$

    $\mathbf{W} \leftarrow \text{UNFOLD}(\mathcal{W}, \text{shape} = \mathbf{d}^{(b)})$ // $\mathbb{R}^{B \times L \times \prod_{k=1}^N d_k^{(b)}}$

    $\mathbf{U}, \mathbf{s}, \mathbf{V} \leftarrow \text{BATCHSVD}(\mathbf{W})$ // $\mathbf{U} \in \mathbb{R}^{B \times L \times R}$ where $R = \min(L, \prod_{k=1}^N d_k^{(b)})$

    $\mathcal{A}^{(i)} \leftarrow \text{STACK}((\text{RESHAPE}(\mathbf{U}_{b,:,r}, \text{shape} = \mathbf{d}^{(a)}) \,|\, b = 1, 2, \ldots B, \ r = 1, 2, \ldots R))$

    $\mathcal{B}^{(i)} \leftarrow \text{STACK}((\text{RESHAPE}(s_k \mathbf{V}_{b,:,r}^\top, \text{shape} = \mathbf{d}^{(b)}) \,|\, b = 1, 2, \ldots B, \ r = 1, 2, \ldots R))$

    $\mathcal{W} \leftarrow \mathcal{B}^{(i)}$

**end**
$\mathcal{A}^{(S)} \leftarrow \mathcal{B}^{S-1}$
**return** $\{\mathcal{A}\}_{i=1}^S$

---

In the first iteration, the tensor being decomposed is the original tensor (i.e., $\mathcal{W}^{(1)} \leftarrow \mathcal{W}$). Whereas in subsequent iterations, intermediate tensors are decomposed. At each iteration, we can convert the problem in equation 6 to the low-rank matrix approximation problem

$$\min_{\substack{\{\mathbf{a}_{r_1 \cdots r_k}^{(k)}, \mathbf{b}_{r_1 \cdots r_k}^{(k)}\} \\ r_j = 1, \cdots R_j, \, j = 1, \ldots, k}} \left\| \mathbf{W}_{r_1 \cdots r_{k-1}}^{(k)} - \sum_{r_k=1}^{R_k} \mathbf{a}_{r_1 \cdots r_k}^{(k)} \mathbf{b}_{r_1 \cdots r_k}^{(k)\top} \right\|_F^2, \tag{7}$$

through reshaping, such that the overall sum of squares is preserved between equation 6 and equation 7. The problem in equation 7 can be readily solved, as it has a well known solution using SVD. The reshaping operations that facilitate this transformation are

$$\mathbf{W}_{r_1 \cdots r_{k-1}}^{(k)} = \text{MAT}(\text{UNFOLD}(\mathcal{W}_{r_1 \cdots r_{k-1}}^{(k)}, \mathbf{d}^{\left(\mathcal{B}_{r_1 \cdots r_k}^{(k)}\right)})), \tag{8}$$

$$\mathbf{a}_{r_1 \cdots r_k}^{(k)} = \text{UNFOLD}(\mathcal{A}_{r_1 \cdots r_k}^{(k)}, \mathbf{d}^{\left(\mathcal{I}_{\mathcal{A}_{r_1 \cdots r_k}^{(k)}}\right)}), \quad \mathbf{b}_{r_1 \cdots r_k}^{(k)} = \text{VEC}(\mathcal{B}_{r_1 \cdots r_k}^{(k)}), \tag{9}$$

where UNFOLD reshapes tensor $\mathcal{W}_{r_1 \cdots r_{k-1}}^{(k)}$ by extracting multidimensional patches of shape $\mathbf{d}_{\mathcal{B}_{r_1, \cdots r_k}^{(k)}}$ from tensor $\mathcal{W}_{r_1 \cdots r_{k-1}}^{(k)}$ in any order, then stacking them along a new first dimension. Vector $\mathbf{d}_{\mathcal{B}}$ denotes a vector describing the shape of a tensor $\mathcal{B}$, $\text{VEC} : \mathbb{R}^{d_1 \times \cdots \times d_N} \to \mathbb{R}^{d_1 \cdots d_N}$ flattens a tensor, $\text{MAT} : \mathbb{R}^{d_1 \times d_2 \times \cdots \times d_N} \to \mathbb{R}^{d_1 \times d_2 \cdots d_N}$ matricizes a tensor and $\mathcal{I}_{\mathcal{A}}$ denotes an identity tensor with the same number of modes as $\mathcal{A}$ and each dimension set to one.

Once each $\mathcal{B}_{r_1 \cdots r_k}^{(k)}$ is obtained by solving equation 7 (and using the inverse of the VEC operation in equation 9), we proceed recursively by setting $\mathcal{W}_{r_1 \cdots r_k}^{(k+1)} \leftarrow \mathcal{B}_{r_1 \cdots r_k}^{(k)}$ and solving the $k+1^{th}$ iteration of equation equation 7. In other words, at the $k^{th}$ iteration, we find Kronecker factors $\mathcal{A}^{(k)}$ and $\mathcal{B}^{(k)}$, where the latter is used in the following iteration. Except in the final iteration (i.e., $k = S - 1$), where the intermediate tensor $\mathcal{B}^{(k)}$ is the solution to the last Kronecker factor $\mathcal{A}^{(S)}$. (See Algorithm 1)

By virtue of the connectivity between all of the Kronecker factors as illustrated in Figure 1a, SeKron can achieve many other commonly used structures, as stated in the following theorem:

**Theorem 2.** *The factorization structure imposed by CP, Tucker, TT and TR when decomposing a given tensor $\mathcal{W} \in \mathbb{R}^{w_1 \times \cdots \times w_N}$ can be achieved using SeKron.*

*Proof.* See Appendix C. $\qquad\square$

### 3.3 CONVOLUTION WITH SEKRON STRUCTURES

In this section, we provide an efficient algorithm for performing a convolution operation using a tensor represented by a sequence of Kronecker factors. Assuming $\mathcal{W}$ is approximated as a sequence

---

**Algorithm 2:** Convolution operation using a sequence of Kronecker factors

---

**Input:** $\{\boldsymbol{\mathcal{A}}^{(i)}\}_{i=1}^{S}, \boldsymbol{\mathcal{A}}^{(i)} \in \mathbb{R}^{r_i \times f_i \times c_i \times Kh_i \times Kw_i} \quad \boldsymbol{\mathcal{X}} \in \mathbb{R}^{N \times C \times H \times W}$

**Output:** $\boldsymbol{\mathcal{X}} \in \mathbb{R}^{N \times \prod_{k=1}^{S} f_k \times H \times W}$

**for** $i \leftarrow S, S-1, \dots, 1$ **do**
    **if** $i == S$ **then**
        $\boldsymbol{\mathcal{X}} \leftarrow \textsc{Conv3d}(\textsc{Unsqueeze}(\boldsymbol{\mathcal{X}}, 1), \textsc{Unsqueeze}(\boldsymbol{\mathcal{A}}^{(i)}, 1))$
        $/\ast \quad \mathbb{R}^{N \times \prod_{k=i+1}^{S} f_k \times r_i f_i \times \prod_{k=1}^{i-1} c_k \times H \times W} \rightarrow \mathbb{R}^{N \times \prod_{k=i}^{S} f_k \times r_{i-1} \times \prod_{k=1}^{i-1} c_k \times H \times W} \qquad \ast/$
        $\boldsymbol{\mathcal{X}} \leftarrow \textsc{Reshape}_1(\boldsymbol{\mathcal{X}})$
    **else**
        $\boldsymbol{\mathcal{X}} \leftarrow \textsc{Conv3d}(\boldsymbol{\mathcal{X}}, \boldsymbol{\mathcal{A}}^{(i)}, \text{groups} = ri)$
        $/\ast \quad \mathbb{R}^{N \times \prod_{k=i+1}^{S} f_k \times r_i f_i \times \prod_{k=1}^{i-1} c_k \times H \times W} \rightarrow \mathbb{R}^{N \times \prod_{k=i}^{S} f_k \times r_i \times \prod_{k=1}^{i-1} c_k \times H \times W} \qquad \ast/$
        $\boldsymbol{\mathcal{X}} \leftarrow \textsc{Reshape}_2(\boldsymbol{\mathcal{X}})$
    **end**
**end**
**return** $\boldsymbol{\mathcal{X}}$

---

of Kronecker products using SeKron, i.e., $\mathcal{W} \approx \widehat{\mathcal{W}}$ and

$$\widehat{\mathcal{W}} = \sum_{r_1=1}^{\widehat{R}_1} \boldsymbol{\mathcal{A}}_{r_1}^{(1)} \otimes \sum_{r_2=1}^{\widehat{R}_2} \boldsymbol{\mathcal{A}}_{r_1 r_2}^{(2)} \otimes \cdots \sum_{r_{S-1}=1}^{\widehat{R}_{S-1}} \boldsymbol{\mathcal{A}}_{r_1 \cdots r_{S-1}}^{(S-1)} \otimes \boldsymbol{\mathcal{A}}_{r_1 \cdots r_{S-1}}^{(S)}, \tag{10}$$

the convolution operation in equation 1 can be re-written as

$$\boldsymbol{\mathcal{Y}}_{fxy} = \sum_{i,j,c=1}^{K_h, K_w, C} \left( \sum_{r_1=1}^{\widehat{R}_1} \boldsymbol{\mathcal{A}}_{r_1}^{(1)} \otimes \cdots \otimes \sum_{r_{S-1}=1}^{\widehat{R}_{S-1}} \boldsymbol{\mathcal{A}}_{r_1 \cdots r_{S-1}}^{(S-1)} \otimes \boldsymbol{\mathcal{A}}_{r_1 \cdots r_{S-1}}^{(S)} \right)_{fcij} \boldsymbol{\mathcal{X}}_{c,i+x,j+y}. \tag{11}$$

Due to the factorization structure of tensor $\widehat{\mathcal{W}}$, the computation in equation 11 can be carried out without its explicit reconstruction. Instead, the projection can be performed using each of the Kronecker factors *independently*. This property is essential to performing efficient convolution operations using SeKron factorizations, and leads to a reduction in both memory and FLOPs at runtime. In practice, this amounts to replacing one large convolution operation (i.e., one with a large convolution tensor) with a sequence of smaller grouped 3D convolutions, as summarized in Algorithm 2.

The ability to avoid reconstruction at runtime when performing a convolution using any SeKron factorization is the result of the following Theorem:

**Theorem 3** (Linear Mappings with Sequences of Kronecker Products). *Any linear mapping using a given tensor* $\mathcal{W}$ *can be written directly in terms of its Kronecker factors* $\boldsymbol{\mathcal{A}}^{(k)} \in \mathbb{R}^{R_1 \times \cdots R_N \times a_1^{(k)} \times \cdots \times a_N^{(k)}}$. *That is:*

$$\mathcal{W}_{i_1 \cdots i_N} \boldsymbol{\mathcal{X}}_{i_1 + z_1, \cdots, i_N + z_N} = \sum_{r_1, \dots r_N}^{R_1, \dots R_k} \boldsymbol{\mathcal{A}}_{r_1 j_1^{(1)} \cdots j_N^{(1)}}^{(1)} \cdots \boldsymbol{\mathcal{A}}_{r_1 \cdots r_{S-1} j_1^{(S)} \cdots j_N^{(S)}}^{(S)} \boldsymbol{\mathcal{X}}_{f(\mathbf{j}_1) + z_1, \cdots, f(\mathbf{j}_N) + z_N}$$

*where* $j_n^{(k)} \in \mathbb{N}$ *is a function of input indices (see Appendix A) and* $f(\mathbf{j}_n) = \sum_{k=1}^{S} j_n^{(k)} \prod_{l=k+1}^{S} a_n^{(l)}$

*Proof.* See Appendix C. $\qquad \square$

Using Theorem 3 we re-write the projection in equation 11 directly in terms of Kronecker factors

$$\boldsymbol{\mathcal{Y}}_{fxy} = \sum_{\mathbf{i},\mathbf{j},\mathbf{c},r_1} \boldsymbol{\mathcal{A}}_{r_1 f_1 c_1 i_1 j_1}^{(1)} \sum_{r_2} \boldsymbol{\mathcal{A}}_{r_1, r_2, f_2, c_2, i_2, j_2}^{(2)} \cdots$$
$$\sum_{r_{S-1}} \boldsymbol{\mathcal{A}}_{r_1 \cdots r_{S-1} f_{N-1} c_{N-1} i_{N-1} j_{N-1}}^{(S-1)} \boldsymbol{\mathcal{A}}_{r_1 \cdots r_{S-1} f_N c_N i_N j_N}^{(S)} \boldsymbol{\mathcal{X}}_{f(\mathbf{c}), f(\mathbf{i})+x, f(\mathbf{j})+y}, \tag{12}$$

where $\mathbf{i} = (i_1, i_2, \dots, i_N)$, $\mathbf{j} = (j_1, j_2, \dots, j_N)$, $\mathbf{c} = (c_1, c_2, \dots, c_N)$ denote vectors containing indices $i_k, j_k, c_k$ that enumerate over positions in tensors $\boldsymbol{\mathcal{A}}^{(k)}$. Finally, exchanging the order of

summation separates the convolution as follows:

$$\boldsymbol{\mathcal{Y}}_{fxy} = \sum_{i_1,j_1,c_1,r_1} \boldsymbol{\mathcal{A}}^{(1)}_{r_1 f_1 c_1 i_1 j_1} \cdots \sum_{i_N,j_N,c_N} \boldsymbol{\mathcal{A}}^{(S)}_{r_1\cdots r_{S-1} f_N c_N i_N j_N} \boldsymbol{\mathcal{X}}_{f(\mathbf{c}),f(\mathbf{i})+x,f(\mathbf{j})+y}. \quad (13)$$

Overall, the projection in equation 13 can be carried out efficiently using a sequence of grouped 3D convolutions with intermediate reshaping operations as described in Algorithm 2. Refer to Appendix C for universal approximation properties of neural networks when using SeKron.

## 3.4 Computational Complexity

In order to decompose a given tensor using our method, the sequence length and the Kronecker factor shapes must be specified. Different selections will lead to different FLOPs, parameters, and latency. Specifically, for the decomposition given by equation 10 for $\widehat{\boldsymbol{\mathcal{W}}} \in \mathbb{R}^{f \times c \times h \times w}$ using factors $\boldsymbol{\mathcal{A}}^{(i)}_{r_1 \cdots r_i} \in \mathbb{R}^{f_i \times c_i \times h_i \times w_i}$, the compression ratio (CR) and FLOPs reduction ratio (FR) are given by

$$\text{CR} = \frac{\prod_{i=1}^{S} f_i c_i h_i w_i}{\sum_{i=1}^{S} \prod_{k=1}^{i} \widehat{R}_k f_i c_i h_i w_i}, \quad \text{FR} = \frac{\prod_{i=1}^{S} f_i c_i h_i w_i}{\sum_{i=1}^{S} \left(\prod_{k=i}^{S} F_k\right)\left(\prod_{k=1}^{i} \widehat{R}_k\right)\left(\prod_{k=1}^{i} c_k\right) h_i w_i}. \quad (14)$$

Applying SeKron to compress DNN models requires a selection strategy for sequence lengths and factor shapes for each layer in a network. We adopt a simple approach that involves selecting configurations that best match a desired CR while also having a lower latency than the original layer being compressed, as FR may not be a good indicator of runtime speedup in practice.

## 4 Experimental Results

To demonstrate the effectiveness of SeKron for model compression, we evaluate different CNN models on both high-level and low-level computer vision tasks. For image classification tasks, we evaluate WideResNet16 (Zagoruyko & Komodakis 2016) and ResNet50 (He et al. 2016) models on CIFAR-10 (Krizhevsky 2009) and ImageNet (Krizhevsky et al. 2012), respectively. For super-resolution task, we evaluate EDSR-8-128 and SRResNet16 trained on DIV2k(Agustsson & Timofte 2017). Lastly, we discuss the latency of our proposed decomposition method. In all experiments we compress convolution layers of pre-trained networks using various compression approaches and then re-train the resulting compressed models. We provide implementation details in Appendix D.

### 4.1 Image Classification Experiments

First, we evaluate SeKron by compressing WideResNet16-8 (Zagoruyko & Komodakis 2016) for image classification on CIFAR-10 and comparing against various approaches. Namely, PCA (Zhang et al. 2016) which imposes that filter responses lie approximately on a low-rank subspace; SVD-Energy (Alvarez & Salzmann 2017) which imposes a low-rank regularization into the training procedure; L-Rank (learned rank selection) (Idelbayev & Carreira-Perpinan 2020) which jointly optimizes over matrix elements and ranks; ALDS (Liebenwein et al. 2021) which provides a global compression framework that finds optimal layer-wise compressions leading to an overall desired global compression rate; TR (Wang et al. 2018a); TT (Novikov et al. 2015) as well as two pruning approaches FT (Li et al. 2017) and PFP (Liebenwein et al. 2020).

Figure 2, shows the CIFAR-10 classification performance drop (i.e., $\Delta$ Top-1) versus compression rates using different methods. SeKron outperforms all other decomposition and pruning methods, at a variety of compression rates. In Table 1 we highlight that at a compression rate of $4\times$ SeKron outperforms all other methods with a small accuracy drop of $-0.51$, whereas the next best decomposition method (omitting rank selection approaches) suffers a $-1.27$ drop in accuracy.

Next, we evaluate SeKron to compress ResNet50 for the image classification task on ImageNet. Table 2 compares our method to other compression approaches. Most notably, SeKron outperforms all decomposition methods, achieving $74.94\%$ Top-1 accuracy which is $\sim 1.1\%$ greater than the second highest accuracy achieved by using TT decomposition. At the same time, SeKron is $3\times$ faster than TT on a single CPU.

Table 1: Performance of compressed WideResNet16-8 using various methods on CIFAR-10

| Model | CR | $\Delta$ Top-1 (%) |
|---|---|---|
| ALDS | 4.0 | $-0.73$ |
| L-Rank | 4.0 | $-3.52$ |
| FT | 4.1 | $-1.50$ |
| PFP | 4.0 | $-0.94$ |
| SVD | 4.0 | $-4.40$ |
| PCA | 4.0 | $-2.08$ |
| SVD Energy | 4.0 | $-1.27$ |
| TT | 4.0 | $-2.86$ |
| TR | 4.0 | $-0.70$ |
| CP | 4.0 | $-3.13$ |
| Tucker | 4.0 | $-1.61$ |
| SeKron (Ours) | 4.1 | $-\textbf{0.51}$ |

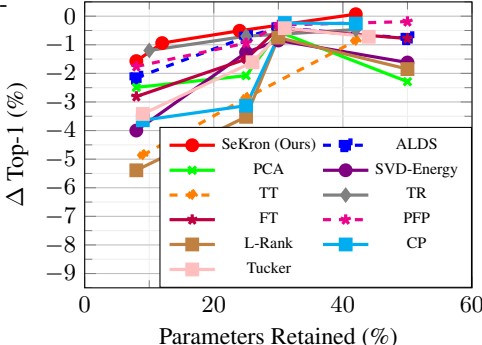

Figure 2: Performance drop of WideResNet16-8 at various compression rates achieved by different methods on CIFAR-10.

Table 2: Performance of ResNet50 using various compression methods measured on ImageNet. [†] indicates models obtained from compressing baselines with different accuracies, for this reason we report accuracy drops of each model with respect to their own baselines as well. The baselines compared are FSNet (Yang et al. 2020), ThiNet (Luo et al. 2017) CP (He et al. 2017) MP (Liu et al. 2019) and Binary Kronecker (Hameed et al. 2022)

| Method | Type | Params (E+6) / CR | FLOPS (E+9) | CPU (ms) | Top-1 / $\Delta$ Top-1 |
|---|---|---|---|---|---|
| FSNet[†] | Other | 13.9 / 1.8 | - | - | 73.11 / $-2.0$ |
| ThiNet[†] | | 12.4 / 2.1 | - | - | 71.01 / $-1.9$ |
| CP[†] | Pruning | - / 2.0 | - | - | 73.30 / $-3.0$ |
| MP[†] | | 10.6 / 2.4 | - | - | 73.40 / $-3.2$ |
| Tensor Ring | | 13.9 / 1.8 | 2.1 | $105 \pm 2$ | 73.30 / $-2.7$ |
| Tensor Train | | 13.3 / 1.9 | 1.9 | $395 \pm 54$ | 73.85 / $-2.1$ |
| Binary Kronecker | Decomposition | 12.0 / 2.1 | - | - | 73.95 / $-2.0$ |
| SeKron $S = 2$ (Ours) | | 12.3 / 2.0 | 2.9 | $125 \pm 3$ | 74.66 / $-1.3$ |
| SeKron $S = 3$ (Ours) | | 13.8 / 1.8 | 2.5 | $133 \pm 4$ | $\textbf{74.94}$ / $-\textbf{1.1}$ |
| Baseline | Uncompressed | 25.5 / 1.0 | 4.10 | $133 \pm 33$ | 75.99 / $-0.0$ |

## 4.2 SUPER-RESOLUTION EXPERIMENTS

In this section we use SeKron to compress SRResNet (Ledig et al. 2017) and EDSR-8-128 (Li et al. 2019). Both networks were trained on DIV2K (Agustsson & Timofte 2017) and valuated on Set5 (Bevilacqua et al. 2012), Set14 (Zeyde et al. 2012), B100(Martin et al. 2001) and Urban100 (Huang et al. 2015). Table 3 presents the performances in terms of PSNR measured on the test images for the models once compressed using SeKron along with the original uncompressed models.

Among model compression methods, Filter Basis Decomposition (FBD) (Li et al. 2019) has been previously shown to achieve state-of-the-art compression on super-resolution CNNs. Therefore, we compare our model compression results with those obtained using FBD as shown in Table 3. We highlight that our approach outperforms FBD, on all test datasets when compressing SRResNet16 at similar compression rates. As this table suggests, when compression rate is increased, FBD results in much lower PSNRs for both EDSR-8-128 and SRResNet16 compared to our proposed SeKron.

## 4.3 CONFIGURING SEKRON CONSIDERING LATENCY AND COMPRESSION RATE

Using the configuration selection strategy proposed in 3.4, we find that a small sequence length ($S$) is limited to few achievable candidate configurations (and consequently compression rates) that do not sacrifice latency. This is illustrated in Figure 3 for $S = 2$ where targeting a CPU latency less than 5 ms and a compression ratio less than $10\times$ leaves only 3 options for compression. In contrast,

Table 3: PSNR (dB) performance of compressed SRResNet16 and EDSR-8-128 ($\times 4$ scaling factor) models using FBD (with basis-64-16) (Li et al. 2019) and our SeKron

| Model | Method | Params (E+6) | CR | Set5 | Set14 | B100 | Urban100 |
|---|---|---|---|---|---|---|---|
| | | | | Dataset | | | |
| SRResNet16 | Baseline | 1.54 | 1.0 | 32.03 | 28.5 | 27.52 | 25.88 |
| | FBD | 0.65 | 2.4 | 31.84 | 28.38 | 27.39 | 25.54 |
| | SeKron | 0.65 | 2.4 | **31.91** | **28.42** | **27.43** | **25.64** |
| | FBD | 0.36 | 4.3 | 31.49 | 28.18 | 27.28 | 25.20 |
| | SeKron | 0.37 | 4.2 | **31.73** | **28.32** | **27.37** | **25.48** |
| EDSR-8-128 | Baseline | 3.70 | 1.0 | 32.13 | 28.55 | 27.55 | 26.02 |
| | FBD | 1.62 | 2.3 | 31.80 | 28.34 | 27.40 | 25.54 |
| | SeKron | 1.50 | 2.5 | 31.79 | 28.34 | 27.39 | 25.52 |
| | FBD | 0.48 | 7.8 | 31.64 | 28.23 | 27.32 | 25.31 |
| | SeKron | 0.47 | 7.8 | **31.77** | **28.32** | **27.38** | **25.46** |

Table 4: CPU latency (ms) for uncompressed (baseline) and compressed SRResNet16 and EDSR-8-128 models using SeKron

| Model | Method | CR | CPU (ms) |
|---|---|---|---|
| SRResNet16 | Baseline | 1.0 | $72 \pm 3$ |
| | SeKron | 2.4 | $70 \pm 5$ |
| | SeKron | 4.2 | $70 \pm 2$ |
| EDSR-8-128 | Baseline | 1.0 | $151 \pm 8$ |
| | SeKron | 2.5 | $124 \pm 4$ |
| | SeKron | 7.8 | $131 \pm 9$ |

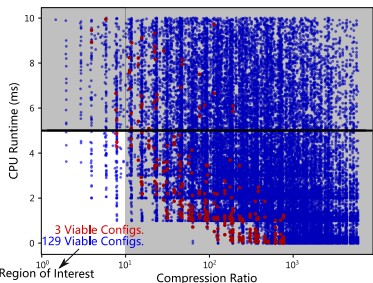

Figure 3: CPU latency for candidate configurations obtained using SeKron on a tensor $\mathcal{W} \in \mathbb{R}^{512 \times 512 \times 3 \times 3}$ with $S = 2$ (red) and $S = 3$ (blue), aiming for a speedup (e.g., $< 5$ ms) and a typical compression rate (e.g., $< 10\times$).

increasing the sequence length to $S = 3$ leads to a wider range of achievable compression rates (i.e., 129 configurations). Despite the flexibility they provide, large sequence lengths lead to an exponentially larger number of candidate configurations and time-consuming generation of all their runtimes. For this reason, unless otherwise stated, we opted to use $S = 3$ in all the above-mentioned experiments as it provided a suitable range of compression rates and a manageable search space.

As an example, in Table 4 we compress EDSR-8-128 using a compression rate of CR $= 2.5\times$, by selecting configurations for each layer that satisfy the desired CR while simultaneously resulting in a speedup. This led to an overall model speedup of 124ms (compressed) vs. 151ms (uncompressed).

## 5 CONCLUSIONS

We introduced SeKron, a tensor decomposition approach using sequences of Kronecker products. SeKron allows for a wide variety of factorization structures to be achieved, while, crucially, sharing the same compression and convolution algorithms. Moreover, SeKron has been shown to generalize popular decomposition methods such as TT, TR, CP and Tucker. Thus, it mitigates the need for time-consuming development of customized convolution algorithms. Unlike other decomposition methods, SeKron is not limited to a single factorization structure, which leads to improved compressions and reduced runtimes on different hardware. Leveraging SeKron's flexibility, we find efficient factorization structures that outperform previous decomposition methods on various image classification and super-resolution tasks.

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

## APPENDIX

### A  SEQUENCE OF KRONECKER PRODUCTS

The Kronecker product between a sequence of factor tensors is given by

$$\left( \boldsymbol{\mathcal{A}}^{(1)} \otimes \cdots \otimes \boldsymbol{\mathcal{A}}^{(S)} \right)_{i_1 \cdots i_N} \triangleq \boldsymbol{\mathcal{A}}^{(1)}_{j_1^{(1)} \cdots j_N^{(1)}} \cdots \boldsymbol{\mathcal{A}}^{(S)}_{j_1^{(S)} \cdots j_N^{(S)}}, \tag{15}$$

where

$$j_n^{(k)} = \begin{cases} i_n - \sum_{t=1}^{k-2} j_n^{(t)} \prod_{l=t+1}^{S} a_n^{(l)} \bmod a_n^{(S)} & k = S, \\ \left\lfloor \dfrac{i_n - \sum_{t=1}^{k-1} j_n^{(t)} \prod_{l=t+1}^{S} a_n^{(l)}}{\prod_{l=k+1}^{S} a_n^{(l)}} \right\rfloor & \text{otherwise}, \end{cases} \tag{16}$$

and $\boldsymbol{\mathcal{A}}^{(k)} \in \mathbb{R}^{a_1^{(k)} \times \cdots \times a_N^{(k)}}$.

### B  ALTERNATIVE EXPANSION DIRECTIONS OF SEKRON

The proposed SeKron structure represents a given tensor $\boldsymbol{\mathcal{W}} \in \mathbb{R}^{w_1 \times \cdots \times w_n}$ using a sequence of Kronecker products as follows:

$$\boldsymbol{\mathcal{W}} = \sum_{r_1=1}^{R_1} \boldsymbol{\mathcal{A}}^{(1)}_{r_1} \otimes \sum_{r_2=1}^{R_2} \boldsymbol{\mathcal{A}}^{(2)}_{r_1 r_2} \otimes \cdots \otimes \sum_{r_{S-1}=1}^{R_{S-1}} \boldsymbol{\mathcal{A}}^{(S-1)}_{r_1 \cdots r_{S-1}} \otimes \boldsymbol{\mathcal{A}}^{(S)}_{r_1 \cdots r_{S-1}}. \tag{4 revisited}$$

While this decomposition structure is obtained by recursively finding the Kronecker decomposition of the right-most tensor, many alternative sequential Kronecker structures can be obtained as illustrated in Figure 4. However, such alternative structures do not fall within our SeKron framework as they cannot make use of our convolution algorithm (Algorithm 2)

### C  THEOREM PROOFS

**Theorem 1** (Tensor Decomposition using a Sequence of Kronecker Products)**.** *Any tensor* $\boldsymbol{\mathcal{W}} \in \mathbb{R}^{w_1 \times \cdots \times w_N}$ *can be represented by a sequence of Kronecker products between* $S \in \mathbb{N}$ *factors:*

$$\boldsymbol{\mathcal{W}} = \sum_{r_1=1}^{R_1} \boldsymbol{\mathcal{A}}^{(1)}_{r_1} \otimes \sum_{r_2=1}^{R_2} \boldsymbol{\mathcal{A}}^{(2)}_{r_1 r_2} \otimes \cdots \otimes \sum_{r_{S-1}=1}^{R_{S-1}} \boldsymbol{\mathcal{A}}^{(S-1)}_{r_1 \cdots r_{S-1}} \otimes \boldsymbol{\mathcal{A}}^{(S)}_{r_1 \cdots r_{S-1}}, \tag{4}$$

*where* $R_i \in \mathbb{N}$ *and* $\boldsymbol{\mathcal{A}}^{(k)} \in \mathbb{R}^{R_1 \times \cdots \times R_k \times a_1^{(k)} \times \cdots \times a_N^{(k)}}$.

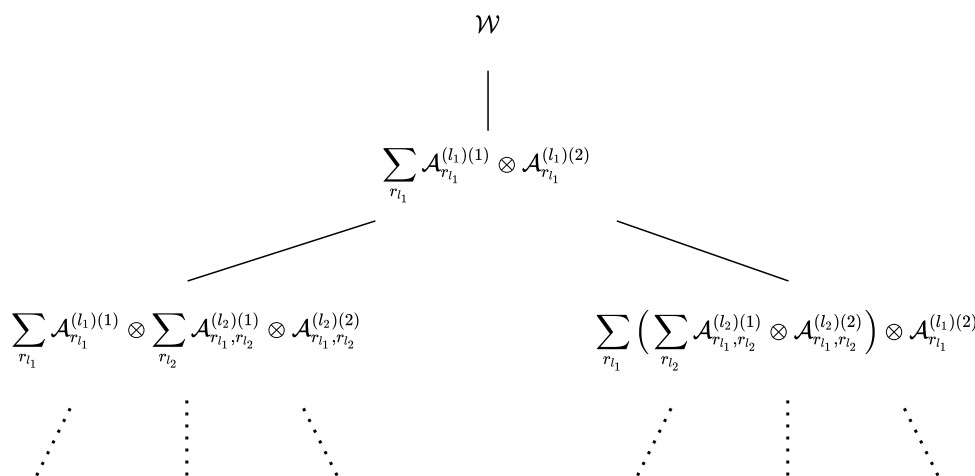

Figure 4: Illustration of alternative expansion directions using sequences of Kronecker products. SeKron structures are those which are leftmost on each level of the tree. Each node is obtained through the decomposition of a single tensor present in its parent node.

*Proof.* First, we define intermediate tensors

$$\boldsymbol{\mathcal{B}}_{r_1\cdots r_k}^{(k)} \triangleq \sum_{r_{k+1}}^{R_{k+1}} \boldsymbol{\mathcal{A}}_{r_1\cdots r_{k+1}}^{(k+1)} \otimes \sum_{r_{k+2}}^{R_{k+2}} \boldsymbol{\mathcal{A}}_{r_1\cdots r_{k+2}}^{(k+2)} \otimes \cdots \otimes \sum_{r_{S-1}}^{R_{S-1}} \boldsymbol{\mathcal{A}}_{r_1\cdots r_{S-1}}^{(S-1)} \otimes \boldsymbol{\mathcal{A}}_{r_1\cdots r_{S-1}}^{(S)} \quad \text{(5 revisited)}$$

Then the reconstruction error can be written as

$$\left\| \boldsymbol{\mathcal{W}}_{r_1\cdots r_{k-1}}^{(k)} - \sum_{r_k=1}^{\widehat{R}_k} \boldsymbol{\mathcal{A}}_{r_1\cdots r_k}^{(k)} \otimes \boldsymbol{\mathcal{B}}_{r_1\cdots r_k}^{(k)} \right\|_{\mathrm{F}}^2 \quad (17)$$

where $\boldsymbol{\mathcal{W}}^{(1)}$ is the initial tensor being decomposed. As described in Section 3.2, using reshaping operations

$$\mathbf{W}_{r_1\cdots r_{k-1}}^{(k)} = \mathrm{MAT}(\mathrm{UNFOLD}(\boldsymbol{\mathcal{W}}_{r_1\cdots r_{k-1}}^{(k)}, \mathbf{d}_{\boldsymbol{\mathcal{B}}_{r_1\cdots r_k}^{(k)}})), \quad \text{(8 revisited)}$$

$$\mathbf{a}_{r_1\cdots r_k}^{(k)} = \mathrm{VEC}(\mathrm{UNFOLD}(\boldsymbol{\mathcal{A}}_{r_1\cdots r_k}^{(k)}, \mathbf{d}_{\boldsymbol{\mathcal{I}}_{\boldsymbol{\mathcal{A}}_{r_1\cdots r_k}^{(k)}}})), \quad \mathbf{b}_{r_1\cdots r_k}^{(k)} = \mathrm{VEC}(\boldsymbol{\mathcal{B}}_{r_1\cdots r_k}^{(k)}), \quad \text{(9 revisited)}$$

that preserve the sum of squares allows us to equivalently write the reconstruction error as

$$\left\| \mathbf{W}_{r_1\cdots r_{k-1}}^{(k)} - \sum_{r_k=1}^{\widehat{R}_k} \mathbf{a}_{r_1\cdots r_k}^{(k)} \mathbf{b}_{r_1\cdots r_k}^{(k)\top} \right\|_{\mathrm{F}}^2. \quad (18)$$

Now consider the singular value decomposition of matrix $\mathbf{W}_{r_1\cdots r_{k-1}}^{(k)}$ and let $\mathbf{u}_{r_1\cdots r_k}^{(k)}, \mathbf{v}_{r_1\cdots r_k}^{(k)}$ denote its left and right singular vectors, respectively (with the right singular vector scaled according to its corresponding singlar value). Set $\mathbf{a}_{r_1\cdots r_k}^{(k)} = \mathbf{u}_{r_k}^{(k)}$ and define and define error terms

$$\delta_{r_1\cdots r_k}^{(k)} = \mathbf{v}_{r_1\cdots r_k}^{(k)} - \mathbf{b}_{r_1\cdots r_k}^{(k)}, \quad \epsilon_{r_1\cdots r_k}^{(k)} = \|\delta_{r_1\cdots r_k}^{(k)}\|. \quad (19)$$

Expanding out equation 18 reveals its recursive form

$$\left\| \mathbf{W}_{r_1 \cdots r_{k-1}}^{(k)} - \sum_{r_k=1}^{\widehat{R}_k} \mathbf{a}_{r_1 \cdots r_k}^{(k)} \mathbf{b}_{r_1 \cdots r_k}^{(k)\top} \right\|_{\mathrm{F}}^2 = \left\| \mathbf{W}_{r_1 \cdots r_{k-1}}^{(k)} - \sum_{r_k=1}^{\widehat{R}_k} \mathbf{a}_{r_1 \cdots r_k}^{(k)} (\mathbf{v}_{r_k}^{(k)} - \delta_{r_1 \cdots r_k}^{(k)})^\top \right\|_{\mathrm{F}}^2 \tag{20}$$

$$= \left\| \mathbf{W}_{r_1 \cdots r_{k-1}}^{(k)} - \sum_{r_k=1}^{\widehat{R}_k} \mathbf{a}_{r_1 \cdots r_k}^{(k)} \mathbf{v}_{r_1 \cdots r_k}^{(k)\top} + \sum_{r_k=1}^{\widehat{R}_k} \mathbf{a}_{r_1 \cdots r_k}^{(k)} \delta_{r_1 \cdots r_k}^{(k)\top} \right\|_{\mathrm{F}}^2 \tag{21}$$

$$\leq \left\| \mathbf{W}_{r_1 \cdots r_{k-1}}^{(k)} - \sum_{r_k=1}^{\widehat{R}_k} \mathbf{a}_{r_1 \cdots r_k}^{(k)} \mathbf{v}_{r_1 \cdots r_k}^{(k)\top} \right\|_{\mathrm{F}}^2 + \sum_{r_k=1}^{\widehat{R}_k} \left\| \mathbf{a}_{r_1 \cdots r_k}^{(k)} \delta_{r_1 \cdots r_k}^{(k)\top} \right\|_{\mathrm{F}}^2 \tag{22}$$

$$\leq \left\| \mathbf{W}_{r_1 \cdots r_{k-1}}^{(k)} - \sum_{r_k=1}^{\widehat{R}_k} \mathbf{a}_{r_1 \cdots r_k}^{(k)} \mathbf{v}_{r_1 \cdots r_k}^{(k)\top} \right\|_{\mathrm{F}}^2 + \sum_{r_k=1}^{\widehat{R}_k} d^{(k)} \epsilon_{r_1 \cdots r_k}^{(k)} \tag{23}$$

$$= \left( \sum_{r_k=\widehat{R}_k+1}^{R_k} \sigma_{r_k}^2 (\mathbf{W}_{r_1 \cdots r_{k-1}}^{(k)}) \right) + \left( \sum_{r_k=1}^{\widehat{R}_k} d^{(k)} \epsilon_{r_1 \cdots r_k}^{(k)} \right) \tag{24}$$

$$= \sum_{r_k=\widehat{R}_k+1}^{R_k} \sigma_{r_k}^2 (\mathbf{W}_{r_1 \cdots r_{k-1}}^{(k)}) + \sum_{r_k=1}^{\widehat{R}_k} d^{(k)} \left\| \mathbf{v}_{r_1 \cdots r_k}^{(k)} - \mathbf{b}_{r_1 \cdots r_k}^{(k)} \right\|_{\mathrm{F}}^2 \tag{25}$$

where $d^{(k)} \in \mathbb{N}$ is the number of dimensions of vector $\mathbf{a}_{r_1 \cdots r_k}^{(k)}$ and $R_k$ is the rank of matrix $\mathbf{W}_{r_1 \cdots r_{k-1}}^{(k)}$, $\sigma_{r_k}(\mathcal{W}_{r_1 \cdots r_{k-1}}^{(k)})$ denotes the $r_k^{\mathrm{th}}$ singular value of tensor $\mathcal{W}_{r_1 \cdots r_{k-1}}^{(k)}$. By reshaping vectors $\mathbf{v}_{r_1 \cdots r_k}^{(k)}, \mathbf{b}_{r_1 \cdots r_k}^{(k)}$ to matrices according to

$$\mathbf{V}_{r_1 \cdots r_k}^{(k)} = \mathrm{M}\mathrm{AT}\left( \mathrm{U}\mathrm{NFOLD}\left( \mathrm{V}\mathrm{EC}^{-1}\left( \mathbf{v}_{r_1 \cdots r_k}^{(k)}, \prod_{s=k+1}^{S} \mathbf{d}^{(s)} \right), \prod_{s=k+2}^{S} \mathbf{d}^{(s)} \right) \right), \tag{26}$$

$$\mathbf{B}_{r_1 \cdots r_k}^{(k)} = \mathrm{M}\mathrm{AT}\left( \mathrm{U}\mathrm{NFOLD}\left( \mathrm{V}\mathrm{EC}^{-1}\left( \mathbf{b}_{r_1 \cdots r_k}^{(k)}, \prod_{s=k+1}^{S} \mathbf{d}^{(s)} \right), \prod_{s=k+2}^{S} \mathbf{d}^{(s)} \right) \right), \tag{27}$$

where $\mathbf{d}^{(s)} = (a_1^{(s)}, \ldots, a_N^{(s)})$ describes the dimensions of the $s^{\mathrm{th}}$ factor, we can re-write equation 25 as

$$\sum_{r_k=\widehat{R}_k+1}^{R_k} \sigma_{r_k}^2 (\mathbf{W}_{r_1 \cdots r_{k-1}}^{(k)}) + \sum_{r_k=1}^{\widehat{R}_k} d^{(k)} \left\| \mathbf{v}_{r_1 \cdots r_k}^{(k)} - \mathbf{b}_{r_1 \cdots r_k}^{(k)} \right\|_{\mathrm{F}}^2 \tag{28}$$

$$= \sum_{r_k=\widehat{R}_k+1}^{R_k} \sigma_{r_k}^2 (\mathbf{W}_{r_1 \cdots r_{k-1}}^{(k)}) + \sum_{r_k=1}^{\widehat{R}_k} d^{(k)} \left\| \mathbf{V}_{r_1 \cdots r_k}^{(k)} - \mathbf{B}_{r_1 \cdots r_k}^{(k)} \right\|_{\mathrm{F}}^2 \tag{29}$$

$$= \sum_{r_k=\widehat{R}_k+1}^{R_k} \sigma_{r_k}^2 (\mathbf{W}_{r_1 \cdots r_{k-1}}^{(k)}) + \sum_{r_k=1}^{\widehat{R}_k} d^{(k)} \left\| \mathbf{V}_{r_1 \cdots r_k}^{(k)} - \sum_{r_{k+1}=1}^{\widehat{R}_{k+1}} \mathbf{a}_{r_1 \cdots r_{k+1}}^{(k+1)} \mathbf{b}_{r_1 \cdots r_{k+1}}^{(k+1)\top} \right\|_{\mathrm{F}}^2 . \tag{30}$$

The last line reveals the recursive nature of the formula (compare with equation 20). Unrolling the recursive formula for $k = 1, \ldots, S-1$, by setting $\mathbf{W}_{r_1 \cdots r_k}^{(k+1)} \leftarrow \mathbf{V}_{r_1 \cdots r_k}^{(k)}$, leads to the following formula for the reconstruction error:

$$\varepsilon_{\mathrm{SeKron}}(\mathbf{W}, \mathbf{r}, \mathbf{D}) = \sum_{r_1=\widehat{R}_1+1}^{R_1} \sigma_{r_1}^2 (\mathbf{W}^{(1)}) + d^{(1)} \sum_{r_1=1}^{\widehat{R}_1} \sum_{r_2=\widehat{R}_2+1}^{R_2} \sigma_{r_2}^2 (\mathbf{W}_{r_1}^{(2)}) + \cdots$$

$$+ d^{(1)} d^{(2)} \cdots d^{(S-2)} \sum_{r_1, r_2, \ldots, r_{S-2}=1}^{\widehat{R}_1, \cdots, \widehat{R}_{S-2}} \sum_{r_{S-1}=\widehat{R}_{S-1}+1}^{R_{S-1}} \sigma_{r_{S-1}}^2 (\mathbf{W}_{r_1 \cdots r_{S-2}}^{(S-1)}) \tag{31}$$

where $\mathbf{r} = (\widehat{R}_1, \ldots, \widehat{R}_{S-1})$ contains the rank values, $\mathbf{D}_s = \mathbf{d}^{(s)}$ contains the Kronecker factor shapes and is referred to as the $\mathbf{Dr}$-SeKron approximation error (note that the dependency of intermediate matrices $\mathbf{W}^{(k)}_{r_1 \cdots r_{k-1}}$ on Kronecker factor shapes $\mathbf{D}$ is implied). Selecting $\widehat{R}_i = R_i \, \forall i$ in equation 31 results in zero reconstruction error. $\qquad \square$

**Theorem 2.** *The factorization structure imposed by CP, Tucker, TT and TR when decomposing a given tensor $\mathcal{W} \in \mathbb{R}^{w_1 \times \cdots \times w_N}$ can be achieved using SeKron.*

*Proof.* The SeKron decomposition of tensor $\mathcal{W}$ is given by

$$\mathcal{W}_{i_1 \cdots i_N} = \sum_{r_1, \ldots, r_S = 1}^{R_1, \cdots, R_S} \mathcal{A}^{(1)}_{r_1 j_1^{(1)} \cdots j_N^{(1)}} \cdots \mathcal{A}^{(S)}_{r_1 \cdots r_{S-1} j_1^{(S)} \cdots j_N^{(S)}} \tag{32}$$

where $\mathcal{A}^{(k)} \in \mathbb{R}^{R_1 \times \cdots \times R_k \times a_1^{(k)} \times \cdots a_N^{(k)}}$ and

$$j_n^{(k)} = \begin{cases} i_n - \sum_{t=1}^{k-2} j_n^{(t)} \prod_{l=t+1}^{S} a_n^{(l)} \bmod a_n^{(S)} & k = S, \\ \left\lfloor \dfrac{i_n - \sum_{t=1}^{k-1} j_n^{(t)} \prod_{l=t+1}^{S} a_n^{(l)}}{\prod_{l=k+1}^{S} a_n^{(l)}} \right\rfloor & \text{otherwise,} \end{cases} \tag{16 revisited}$$

The CP decomposition of tensor $\mathcal{W}$ in scalar form is

$$\mathcal{W}_{i_1 \cdots i_N} = \sum_{r=1}^{R^{(\mathrm{CP})}} \mathcal{A}^{(\mathrm{CP}_1)}_{r i_1} \cdots \mathcal{A}^{(\mathrm{CP}_N)}_{r i_N} \tag{33}$$

where $\mathcal{A}^{(\mathrm{CP}_k)} \in \mathbb{R}^{R^{(\mathrm{CP})} \times w_k}$. Configuring the SeKron decomposition in equation 32 such that $S = N$; $R_1 = R^{(\mathrm{CP})}$; $R_2, \ldots, R_N = 1$ and $a_n^{(n)} = w_n$ for $n = 1, \ldots, N$ leads to the equivalent form

$$\mathcal{W}_{i_1 \cdots i_N} = \sum_{r_1 = 1}^{R^{(\mathrm{CP})}} \mathcal{A}^{(1)}_{r_1 i_1 1 \cdots 1} \cdots \mathcal{A}^{(N)}_{r, 1 \cdots 1 i_N}. \tag{34}$$

The Tucker decomposition of tensor $\mathcal{W}$ is given by

$$\mathcal{W}_{i_1 \cdots i_N} = \sum_{r_1 = 1, \ldots, r_N}^{R_1^{(\mathrm{T})}, \ldots R_N^{(\mathrm{T})}} \mathcal{G}_{r_1 \cdots r_N} \mathcal{A}^{(\mathrm{T}_1)}_{i_1 r_1} \cdots \mathcal{A}^{(\mathrm{T}_N)}_{i_N r_N} \tag{35}$$

where $\mathcal{G} \in \mathbb{R}^{R_1^{(\mathrm{T})} \times \cdots \times R_N^{(\mathrm{T})}}$ and $\mathcal{A}^{(\mathrm{T}_k)} \in \mathbb{R}^{w_k \times R_k^{(\mathrm{T})}}$. The SeKron decomposition of tensor $\mathcal{W}$, with $S = N+1$, $R_n = R_n^{(T)}$ and $a_n^{(n)} = w_n$ for $n = 1, \ldots, N$ yields

$$\mathcal{W}_{i_1 \cdots i_N} = \sum_{r_1, \ldots, r_N = 1}^{R_1^{(\mathrm{T})}, \cdots, R_N^{(\mathrm{T})}} \mathcal{A}^{(1)}_{r_1 i_1 1 \cdots 1} \cdots \mathcal{A}^{(N)}_{r_1 \cdots r_N 1 \cdots 1 i_N} \mathcal{A}^{(N+1)}_{r_1 \cdots r_N 1 \cdots 1}, \tag{36}$$

which is equivalent to equation 35 in the special case where there are nullity constraints on some elements in the Kronecker factors, such that for $k = 2, \ldots, N$

$$\mathcal{A}^{(k)}_{r_1 \cdots r_k 1 \cdots 1 i_k 1 \cdots 1} = 0 \quad \text{when} \quad r_j \in \{x \in \mathbb{N} \mid x \le R_j^{(\mathrm{T})}, \ x \ne R_j^{(\mathrm{T}^*)}\} \quad j = 1, \ldots, k-1 \tag{37}$$

for any choice of $R_j^{(\mathrm{T}^*)} \in \{x \in \mathbb{N} \mid x \le R_j^{(\mathrm{T})}\}$. The Tensor Ring (TR) decomposition of $\mathcal{W}$ is given by

$$\mathcal{W}_{i_1 \cdots i_N} = \sum_{r_1, \ldots, r_N = 1}^{R_1^{(\mathrm{TR})}, \ldots R_N^{(\mathrm{TR})}} \mathcal{A}^{(\mathrm{TR}_1)}_{i_1 r_1 r_2} \cdots \mathcal{A}^{(\mathrm{TR}_N)}_{i_N r_N r_{N+1}} \tag{38}$$

where $\mathcal{A}^{(\mathrm{TR}_k)} \in \mathbb{R}^{w_k \times R_k^{(\mathrm{TR})} \times R_{k+1}^{(\mathrm{TR})}}$, and $R_1^{(\mathrm{TR})} = R_{N+1}^{(\mathrm{TR})}$. As the Tensor Train decomposition can be viewed as a special case of the Tensor Ring decomposition (with $R_1^{(\mathrm{TR})} = R_{N+1}^{(\mathrm{TR})} = 1$), it suffices to

show that SeKron generalizes Tensor Ring. The SeKron decomposition of tensor $\mathcal{W}$, with $S = N+1$; $R_k = R_k^{(\text{TR})}$ for $k = 1, \ldots, N-1$ and $a_n^{(n+1)} = w_n$ for $n = 1, \ldots, N$ leads to

$$\mathcal{W}_{i_1 \cdots i_N} = \sum_{r_1, \ldots, r_N = 1}^{R_1^{(\text{TR})}, \ldots, R_N^{(\text{TR})}} \mathcal{A}_{r_1 1 \cdots 1}^{(1)} \mathcal{A}_{r_1 r_2 i_1 1 \cdots 1}^{(2)} \cdots \mathcal{A}_{r_1 \cdots r_{N+1} 1 \cdots 1 i_N}^{(N+1)}, \tag{39}$$

which is equivalent to equation 38 in the special case where some elements in the Kronecker factors are constrained, such that all elements in tensor $\mathcal{A}^{(1)}$ are constrained to one and

$$\mathcal{A}_{r_1 \cdots r_k 1 \cdots 1 i_k 1 \cdots 1}^{(k)} = 0 \quad \forall r_j \in \{x \in \mathbb{N} \mid x \leq R_j^{(\text{TR})}, \ x \neq R_j^{(\text{TR}^*)}\} \tag{40}$$

for

$$j = \begin{cases} 1, \ldots, k-2 & k = 2, \ldots, N \\ 2, \ldots, k-1 & k = N+1 \end{cases} \tag{41}$$

for any choice of $R_j^{(\text{TR}^*)} \in \{x \in \mathbb{N} \mid x \leq R_j^{(\text{TR})}\}$. $\qquad\square$

**Theorem 3** (Linear Mappings with Sequences of Kronecker Products). *Any linear mapping using a given tensor $\mathcal{W}$ can be written directly in terms of its Kronecker factors $\mathcal{A}^{(k)} \in \mathbb{R}^{R_1 \times \cdots R_N \times a_1^{(k)} \times \cdots \times a_N^{(k)}}$. That is:*

$$\mathcal{W}_{i_1 \cdots i_N} \mathcal{X}_{i_1 + z_1, \cdots, i_N + z_N} = \sum_{r_1, \ldots r_N}^{R_1, \ldots R_k} \mathcal{A}_{r_1 j_1^{(1)} \cdots j_N^{(1)}}^{(1)} \cdots \mathcal{A}_{r_1 \cdots r_{S-1} j_1^{(S)} \cdots j_N^{(S)}}^{(S)} \mathcal{X}_{f(\mathbf{j}_1) + z_1, \cdots, f(\mathbf{j}_N) + z_N}$$

*where $j_n^{(k)} \in \mathbb{N}$ is a function of input indices (see Appendix A) and $f(\mathbf{j}_n) = \sum_{k=1}^{S} j_n^{(k)} \prod_{l=k+1}^{S} a_n^{(l)}$*

*Proof.* First we bring out the summations in the SeKron representaion of $\mathcal{W}$

$$\mathcal{W} = \sum_{r_1}^{R_1} \mathcal{A}_{r_1}^{(1)} \otimes \sum_{r_2}^{R_2} \mathcal{A}_{r_1 r_2}^{(2)} \otimes \cdots \otimes \sum_{r_{S-1}}^{R_{S-1}} \mathcal{A}_{r_1 \cdots r_{S-1}}^{(S-1)} \otimes \mathcal{A}_{r_1 \cdots r_{S-1}}^{(S)}, \tag{4 revisited}$$

such that

$$\mathcal{W} = \sum_{r_1, \ldots, r_S = 1}^{R_1, \cdots, R_{S-1}} \mathcal{A}_{r_1}^{(1)} \otimes \cdots \otimes \mathcal{A}_{r_1 r_2 \cdots r_{S-1}}^{(S)}. \tag{42}$$

Then, using the scalar form definition of sequences of kronecker products in equation 16

$$j_n^{(k)} = \begin{cases} i_n - \sum_{t=1}^{k-2} j_n^{(t)} \prod_{l=t+1}^{S} a_n^{(l)} \bmod a_n^{(S)} & k = S, \\ \left\lfloor \dfrac{i_n - \sum_{t=1}^{k-1} j_n^{(t)} \prod_{l=t+1}^{S} a_n^{(l)}}{\prod_{l=k+1}^{S} a_n^{(l)}} \right\rfloor & \text{otherwise,} \end{cases} \tag{16 revisited}$$

allows us to re-write equation 42 in scalar form as

$$\mathcal{W}_{i_1 \cdots i_N} = \sum_{r_1 \cdots r_S = 1}^{R_1} \mathcal{A}_{r_1 j_1^{(1)} \cdots j_N^{(1)}}^{(1)} \cdots \mathcal{A}_{r_1 \cdots r_{S-1} j_1^{(S)} \cdots j_N^{(S)}}^{(S)} \tag{43}$$

As the $j_n^{(k)}$ terms decompose $i_n$ into an integer weighted sum, we can recover $i_n$ using

$$i_n = f(\mathbf{j}_n) \triangleq \sum_{k=1}^{S} j_n^{(k)} \prod_{l=k+1}^{S} a_n^{(l)}, \tag{44}$$

where $\mathbf{j}_n = (j_n^{(1)}, \ldots, j_n^{(S)})$. Thus, we can write

$$\mathcal{X}_{i_1 + z_1, \cdots i_N + z_N} = \mathcal{X}_{f(\mathbf{j}_1) + z_1, \cdots f(\mathbf{j}_N) + z_N}. \tag{45}$$

Finally, combining equations equation 43 and equation 45 leads to

$$\mathcal{W}_{i_1 \cdots i_N} \mathcal{X}_{i_1 + z_1, \cdots, i_N + z_N} = \sum_{r_1, \ldots r_N}^{R_1, \ldots R_k} \mathcal{A}_{r_1 j_1^{(1)} \cdots j_N^{(1)}}^{(1)} \cdots \mathcal{A}_{r_1 \cdots r_{S-1} j_1^{(S)} \cdots j_N^{(S)}}^{(S)} \mathcal{X}_{f(\mathbf{j}_1) + z_1, \cdots, f(\mathbf{j}_N) + z_N}$$

$$\square$$

**Theorem 4.** *(Universal approximation via shallow SeKron networks) Any shallow SeKron factorized neural network $\hat{f}^{(s)}$ with an L-Lipschitz activation function a, is dense in the class of continuous functions $C(X)$ for any compact subset $X$ of $\mathbb{R}^d$*

*Proof.* Let $\hat{f}$ denote a shallow neural network, and $f \in C(X)$. Then,

$$\left\| f - \hat{f}^{(s)} \right\|_2^2 \triangleq \int_X \left( f(x) - \hat{f}^{(s)}(x) \right)^2 d\mu \tag{46}$$

$$= \int_X \left( f(x) - \hat{f}(x) \right)^2 d\mu \tag{47}$$

$$+ \int_X \left( \hat{f}(x) - \hat{f}^{(s)}(x) \right)^2 d\mu \tag{48}$$

$$+ 2 \int_X \left( f(x) - \hat{f}(x) \right) \left( \hat{f}(x) - \hat{f}^{(s)}(x) \right) d\mu \tag{49}$$

According to Hornik (1991), equation 47 is dense in $C(X)$; therefore, it suffices to show that equation 48 is bounded as well.

$$\int_X \left( \hat{f}(x) - \hat{f}^{(s)}(x) \right)^2 d\mu = \int_X \left( \mathbf{w}^\top \mathbf{a}(\mathbf{W}x) - \mathbf{w}^\top \mathbf{a}(\mathbf{W}^{(s)}x) \right)^2 d\mu \tag{50}$$

$$\leq L \|\mathbf{w}\|_2^2 \|X\|_2^2 \varepsilon_{\text{SeKron}}(\mathbf{W}, \mathbf{r}, \mathbf{D}) \tag{51}$$

where $\varepsilon$ denotes the $\mathbf{Dr}$-SeKron approximation error as in equation 31, with matrix $\mathbf{D}$ and vector $\mathbf{r}$ describing the shapes of the Kronecker factors the ranks used in the SeKron decomposition of $\mathbf{W}$, respectively. $\qquad\square$

## D  IMPLEMENTATION DETAILS

In all of our experiments we use 4 NVIDIA Tesla V100 SXM2 32 GB GPUs during training and evaluate run time on a single core of Intel(R) Xeon(R) Gold 6148 CPU @ 2.40GHz.

### D.1  IMAGENET EXPERIMENTS

We train all models using stochastic gradient descent for 100 epochs using a batch size of 256. The learning rate is initially set to 0.1 and reduced by a factor of $10\times$ at epochs number 30, 60 and 90. We also use a 0.0001 weight decay.

### D.2  CIFAR-10 EXPERIMENTS

We train all models using using stochastic gradient descent for 200 epochs using a batch size of 128. The learning rate is initially set to 0.1 and is reduced by a factor of $5\times$ at epochs number 60, 120 and 160. We use nestrov momentum set to 0.9 and weight decay set to 0.0005.

### D.3  DIV2K

We train all models using using the ADAM optimizer for 300 epochs using a batch size of 16. The optimizer's learning rate is set to 0.0001 and $\beta_1, \beta_2$ are set to $0.9, 0.999$ respectively.

