# OpenReview forum: "SeKron: A Decomposition Method Supporting Many Factorization Structures"
_ICLR.cc/2023/Conference — Submitted to ICLR 2023_

### Official Review · Reviewer_Pd7U · 2022-10-22

**Confidence:** 4
**Correctness:** 2
**Technical Novelty And Significance:** 2
**Empirical Novelty And Significance:** 3
**Recommendation:** 1

**Clarity, Quality, Novelty And Reproducibility:**

--- Clarity/Quality ---

The paper is unclear in many respects. In addition to the confusing points listed under weaknesses, some further points follow below.
- You should be more specific about the difference between high- and low-level computer vision tasks in the introduction/throughout the paper.
- In Fig 1 (a), the tensors in SeKron are incorrectly numbered.
- What are the sizes of the factor tensors used in the experiments? This is particularly relevant to state since there is a requirement that the dimensions of the factor tensors factorize the corresponding dimensions of the full tensor.

--- Novelty ---

It's unclear how the paper is different from previous work like that by Hameed et al. (2022).

--- Reproducibility ---

In its current form, it's not reproducible due to the lack of details, poor notation, and lack of code.

**Details Of Ethics Concerns:**

It seems very likely that the authors have reduced white spaces in their paper in order to squeeze in more material within the 9-page limit.

**Strength And Weaknesses:**

--- Strengths ---

S1. Sections 1-2 provide a nice introduction to the area. The overview in Section 2 is particularly good. I also appreciated the clear introduction to CNN layers at the start of Section 3.1.

--- Weaknesses ---

W1. The authors have clearly reduced whitespace throughout the paper; equations are crammed together, captions are too close to the figures. This by itself is grounds for rejection since it effectively violates the 9-page paper limit.

W2. An important weakness that is not mentioned anywhere is that the factors $A^{(k)}$ in Eq (8) must have dimensions that factorize the dimensions of $W$. For example, they must satisfy $\prod_{k=1}^S a_j^{(k)} = w_j$. So what is hailed as *greater flexibility* of the proposed model in the caption of Fig 1 is in fact a *limitation*. For example, if the dimensions of $W$ are prime numbers, then for each mode of $W$, only a single tensor $A^{(k)}$ can have a non-singleton dimension in that same mode. This may be fixable with appropriate zero padding, but this has to at least be discussed and highlighted in the paper.

W3. The 2nd point in the list of contributions in Sec 1 claims that the paper provides a means of finding the *best approximation* in the proposed format. In fact, it is easy to see that this claim is likely to be false: The decomposition corresponds to a difficult non-convex optimization problem, and it is therefore unlikely that a simple algorithm with a finite number of steps could solve it optimally.

W4. SeKron is claimed to generalize various other decompositions. But it is not clear that the proposed algorithm could ever reproduce those decompositions. For example, since there is no SVD-based algorithm for CP decomposition, I strongly suspect that the proposed algorithm (which is SVD-based) cannot recreate the decomposition that, say, an alternating least squares based approach for CP decomposition would achieve.

W5. The paper is unclear and poor notation is used in multiple places. For examples:
- Subscripts are sometimes used to denote indices (e.g., Eq (5)), sometimes to denote sequences of tensors (e.g., Eqs (7), (8)), and sometimes used to denote both at the same time (e.g., Thm 3, Eq (35))! This is very confusing.
- It is unclear how Eq (7) follows from Eq (5). The confusing indices exacerbate this.
- In Thm 1, $A^{(k)}$ are tensors, so it's unclear what you mean by "$R_i$ are ranks of intermediate matrices".
- In Alg 1, you apply SVD to a 3-way tensors. This operation is not defined. If you mean batched SVD, you need to specify that.
- The $W_{r_1 \cdots r_{k-1}}^{(k)}$ tensors in Eq (10) haven't been defined.
- The definition of Unfold below Eq (13) is ambiguous. Similarly, you say that Mat reformulates a tensor to a matrix, but list the output space as $R^{d_1 \cdots d_N}$, i.e., indicating that the output is a vector.
- Below Eq (15) you discuss "projection". This is not an appropriate term to use, since these aren't projections; projection is a term with a specific meaning in linear algebra.
- In Eq (16), the $r_k$ indices appear on the right-hand side but not on the left-hand side.

**Summary Of The Paper:**

This paper considers compression of CNN weight tensors by using a kind of tensor decomposition. The purpose of this is to reduce the number of parameters and speed up the convolution computation.

It is quite clear that the authors have compressed various white spaces in the paper (e.g., around equations, in captions, around floats). This, by itself, is grounds for rejection. In addition to this, the paper is confusing in many places, and key weaknesses of the approach aren't discussed.

### Update after rebuttal ###

I leave my score as it is due, to the following reasons:
- The original submission violated the page limit due to substantial alterations of white spaces.
- The paper doesn't sufficiently address the limitations with how dimensions of the factors have to be chosen. For a method like this to be useful, I think it should be relatively insensitive to the data tensor dimensions. For example, if the tensor to be decomposed is of size $1024 \times 1024 \times 1024$, then SeKron has a great deal of flexibility. In particular, since $1024=2^{10}$, the number of factors $S$ can be chosen to be anywhere from 1 to 10 and the sizes $a_1^{(k)}, a_2^{(k)}, a_3^{(k)}$ for $k=1,\ldots,S$ can satisfy the requirement $\prod_{k=1}^S a_i^{(k)} = 1024$ in many different ways. But if this is changed just slightly to a tensor of size $1021 \times 1021 \times 1021$, then all of a sudden there is no flexibility at all. In particular, since 1021 is a prime number, you have to choose $S=1$ and $a_i^{(1)} = 1021$ for each $i =1,2,3$. This sensitivity in the behavior of the model to the input tensor size is, in my opinion, a substantial weakness. This could potentially be alleviated with appropriate zero padding. But this isn't done in the paper, and the authors don't even acknowledge the weakness of their current model.

**Summary Of The Review:**

The authors have reduced white spaces of the paper, effectively violating the 9-page limit. In addition to this, there are issues with notation  that make the paper hard to follow. Important limitations haven't been discussed.

---

> ### Author Response · Authors · 2022-11-18
> **Typesetting, Factorization constraint, deriving structures and not numerical solutions**
>
> **A1:**
>
> We would like to apologize for this inconvenience, and we have reset the whitespaces to their defaults. (Please see the updated PDF version).
>
>
> **A2:** Satisfying $\prod_{k=1}^{S} a_j^{k} = w_j$
>
>
> We are not sure what is meant by our method being ``limited'' in your comment, as it is similar to saying that the SVD factorization of a matrix is limited because in the matrix equation A=BC the number of columns of B must be equal to the rows of C. Our factorization method, like any other factorization method, must satisfy dimension constraints indeed.
>
> In the case of a tensor $\mathcal{W}$ with prime dimensions we have just as much flexibility as any of the other decomposition methods (combined). So we do indeed have greater flexibility than any of the other decomposition approaches listed.
>
> For example, for a given tensor $\mathcal{W}$ with prime dimensions a decomposition approach such as CP will have to choose factors whose dimensions are prime (with an arbitrary inner rank dimension), call the set of all possible CP factorizations F_CP. The same would be true for using the tucker decomposition, resulting in a set of possible factorizations F_Tucker. Now because of the flexibility of the SeKron representation, we can represent tensor $\mathcal{W}$ using a set of factorization F_SeKron $\supseteq$ F_CP $\cup$ F_Tucker. In fact it is a superset of the union of structures achieved by other approaches as well such as Tensor Ring and Tensor Train. This is what is meant by greater flexibility. That is, SeKron is more flexible that TT, CP, Tucker, TR combined.
>
> Finally, we appreciate you pointing out that we should mention the factorization constraint (i.e. $\prod_{k=1}^{S}a_j^{k} = w_j$) and have included this in the updated version. (See paragraph above eq.2)
>
> **A3:**
>
> Thank you for bringing this to our attention. This was a typo in the introduction, that has been corrected (see revised version of the pdf). However, we would like to point out that in the body of the paper we did previously state that we are providing a quasi-optimal solution (Theorem 1). As is common for solvers in tensor decomposition methods such as TT-SVD[1], we provide a solution method that can achieve arbitrarily small error but does not necessarily provide the global minimum.
>
> **A4:**
>
> We do not claim that we can re-create the numerical solution obtained by other solvers such as  Alternating Least Squares. Our claim in Theorem 2 is that we can derive the **structures** of those other methods (CP, Tucker Tensor Train and Tensor Ring), not the actual values obtained by whichever respective solvers are used. To clarify this point further we have changed the wording of Theorem 2 to say "factorization" instead of "decomposition".
>
> **A5:**
>
> While resetting the white-space we have moved two equations to the appendix, as as result the equation numbers have changed slightly. In our answer we will use the new equation numbers but include the old equation number in brackets to make it easier to follow.
>
> 1. In all cases the subscripts denote indices. In Eq. 15 (5) the subscripts are indexing spatial dimensions and in Eq 3 (7) and Eq 4 (8), they are indexing rank dimensions. Subscripts are never used to denote a sequence, only superscripts are used for that. Perhaps the indexing of rank dimensions was not clear previously, so we have written the dimension of tensor A out explicitly above Eq 3 (7) to make it more clear. As well as fixed a typo regarding the dimensionality of A below Eq 4 (8) and below Eq. 32 (35).
>
> 2. To see how Eq. 3 (7) follows from Eq 15 (5) consider the special case of $R_i = 1$ for all $i$. We have added brackets to this equation in case that is what caused confusion in the updated pdf and also link to it here:
> https://drive.google.com/file/d/1d8bHwtuWogL9WBkXmT9laSMKmR1Cx0aP/view?usp=sharing
>
> 3. We have removed the "rank of intermediate matrices" and made the theorem statement simpler:
> https://drive.google.com/file/d/125H6eymndBB9z4p5kV1TmKtqrr_D-g0S/view?usp=share_link
>
> 4. Yes this is indeed Batched SVD and we re-named it accordingly: https://drive.google.com/file/d/1gOaOZWJkaPd0VzTIMZVskfLfiiT_y8VV/view?usp=sharing
>
> Please see next comment for our response regarding the last few typos.
>
> [1] Ivan V Oseledets. Tensor-train decomposition. SIAM Journal on Scientific Computing, 33(5):
> 2295–2317, 2011.

---

> > ### Author Response · Authors · 2022-11-18
> > **Addressing the last few typos**
> >
> > 5. The $\mathcal{W}^{(k)}$ tensors are indeed defined. Here the $\mathcal{W}^{(k)}$ represents tensors that are to be decomposed recursively. As we are describing a recursive process we define the tensor of the first iteration $\mathcal{W}^{(1)}$ in the sentence immediately after it is written (https://drive.google.com/file/d/1i083oVCxTR8FAoTWTjqAW0AovQLpaK2d/view?usp=sharing). We then proceed with the next steps of the recursive process until we have constructed the next intermediate tensor $\mathcal{B}^{(k)}$ in the iteration and then complete the definition for the rest of the tensors W^{k+1} (which are copied over from intermediate tensors $\mathcal{B}^{(k)}$: https://drive.google.com/file/d/1RQl1ajmVqY21-nTQCNnHLhpbgsRODg0h/view?usp=sharing).
> >
> > 6. (a) The unfold operation is defined up to an arbitrary ordering of patch extraction, the reason we do not define any such ordering is because the rest of our analysis is valid for any chosen ordering. We have an additional sentence clarifying that the unfold stacks the extracted tensors (patches) along a new first dimension. (b) The typo regarding the MAT operator has been fixed.
> >
> > 7. We have removed the term "projection" here and replaced it with "computation".
> >
> > 8.  The $r_k$ terms are summation indices, there was a typo here missing the summation sign. We have fixed this typo: https://drive.google.com/file/d/1WTDdYVkadL2gsnFjDRBHMLkESVvn5vaU/view?usp=sharing
> >
> > We would like to thank you for pointing out some of these typos

---

> ### Author Response · Authors · 2022-11-23
> **Response to Reviewer Pd7U's "Update after rebuttal"**
>
> Thank you for providing an update. Regarding your 1st reason, we fixed those typesetting/formatting issues and the revised version fits well in the ICLR format.
>
> Regarding your 2nd reason for the strong reject decision:
> If we understand your comment correctly, you are implying that the sequence length is limited to $S=1$ when the dimensions are prime and therefore our method has no flexibility in this setting ("...since 1021 is a prime number, you have to choose $S=1$..."). This is in fact **not true** and we would like to clarify this point with the following example. Let $W$ be a tensor with prime dimensions of size $1021 \times 1021 \times 1021$. Then, we can represent it with a SeKron decomposition of arbitrary length (here with S=3 for instance) as follows:
>
> $W = \sum_{r_1}^{R_1} A_{r_1}^{(1)} \otimes  \sum_{r_2}^{R_2} A_{r_1 r_2}^{(2)} \otimes  A_{r_1 r_2}^{(3)}$
>
> where $A^{(1)} \in \mathbf{R}^{R_1 \times 1021 \times 1 \times 1}$, $A^{(2)} \in \mathbf{R}^{R_1 \times R_2 \times 1 \times 1021 \times 1}$ and $A^{(3)} \in \mathbf{R}^{R_1 \times R_2 \times 1 \times 1 \times 1021}$. So we still have flexibility in choosing the sequence length and rank dimensions indeed, and are not limited to $S=1$. Compare this for example with the CP decomposition given by
>
> $W = \sum_{r}^{R} A_{r}^{(1)} \otimes A_{r}^{(2)} \otimes  A_{r}^{(3)}$
>
> where $A^{(1)}, A^{(2)}, A^{(3)} \in \mathbf{R}^{R\times 1021}$
>
> In both cases, SeKron and CP, the factorizations have flexibility in choosing rank dimensions but must satisfy the constraint on factorizing the 1021 dimension. This is indeed the case for other methods as well such as TT, TR and Tucker. Moreover, it is easy to see the SeKron factorization generalizes the CP structure (choose $R_2 = 1$). In the paper we show SeKron structures can also achieve the structures of TT, TR and Tucker. Thus making it more flexible than any one of those methods.

---

> > ### Comment · Reviewer_Pd7U · 2022-11-23
> > **Issue with sensitivity to input dimensions remains**
> >
> > I see what you mean; indeed you can choose $S$ to be greater than one. But the issue that the model is much more restricted when the dimensions are prime than when they are a, e.g., a power of 2 remains (due to the requirement $\prod_{k=1}^S a_j^{(k)} = w_j$). To me it's a fundamental issue that a minor change in the input tensor dimensions vastly impacts the flexibility of the model.

---

> > > ### Author Response · Authors · 2022-11-23
> > > **Response to sensitivity to input dimensions**
> > >
> > > Thank you for your comment. The reason we do not refer to this as a limitation is because it is a basic requirement of any factorization method. The Singular value decomposition of a matrix for instance, $A = U\Sigma V^\top$, requires that the rows of $U$ equal the rows of A (and columns of $V^\top$ equal to columns of $A$), with only the inner rank term being a choice during compression. The point we're making is that this is a **common requirement**  present in any factorization method SVD, Eigen decomposition, Tensor Train, Tensor Ring, CP and Tucker. We are no different in needing to satisfy this common requirement. Therefore, we do not view it as a limitation of our proposed method.
> > >
> > > In the final version of the paper, we will highlight that all existing decomposition methods have a lower flexibility in case of factorizing tensors whose dimensions are prime.

---

### Official Review · Reviewer_Qck6 · 2022-10-25

**Confidence:** 4
**Correctness:** 3
**Technical Novelty And Significance:** 3
**Empirical Novelty And Significance:** Not applicable
**Recommendation:** 8

**Clarity, Quality, Novelty And Reproducibility:**

Another thing that has been brought up in the previous review (mentioned by other reviewers) is the position of this paper in the context of tensor decomposition literature. It has been criticized that the SVD-based approximation technique is not new and this might cause confusion in this context. The claim that TT / TR are special cases of SeKron (shown by a constructive proof) was also criticized in that review. These points were not clearly addressed in this version. I was personally somewhat against these criticisms in that they may be valid in terms of tensor decomposition literature, but they entirely miss the point (and contribution) of this paper, i.e., proposing a flexible decomposition structure that is efficient for neural-net compression. Existing tensor decomposition may also be used for neural-net compression, but definitely not with the flexibility SeKron has. The decomposition format as well as the solver is important in the tensor decomposition context, but for neural-net compression, I believe that the former can be much more important.

As such, I suggest the authors limit the scope of the paper and more clearly describe the contributions. The authors might want to tone down regarding the generality of SeKron but rather emphasize the flexibility that suits the purpose of neural-net compression. In other words, SeKron may not be new in many aspects of the existing tensor decomposition context, except for the flexible decomposition format. The successive SVD-based approximation can be one of these aspects, i.e., it is not entirely new in the literature, but it is rather the simplest example solver that can calculate SeKron easily. I also suggest limiting the discussion regarding the comparison of formats between different decomposition structures strictly to the neural-net compression perspective. Accordingly, I give a review score of 6 and will determine the final score after seeing the rebuttal.

I believe that the experiments in the paper are reproducible.

**Strength And Weaknesses:**

The main idea is quite simple and intuitive (i.e., representing any possible form of tensor products with a series of Kronecker products), but it is definitely a worthy contribution. What the authors have proposed is basically a general tensor decomposition structure that is easy to manipulate. I was personally surprised by the fact that nobody has attempted this (which now seems to be a natural choice for network compression) until this moment, and I also think that this reflects the importance of this contribution. What existing tensor decompositions have focused on was mostly things like having good algebraic properties or finding the best approximation with respect to some criteria. On the other hand, the proposed method might not be particularly better (or might be even worse in some aspects) for the above properties, but it is definitely more flexible in determining the shapes of the factors. For network compression purposes, this can be a better choice.

In fact, I was one of the reviewers of this paper in another venue. I see that some of my previous concerns have been addressed in this paper. I asked about the binary Kronecker decomposition in the experimental comparison, and I see that it has been added in this version. However, the "direction of expansion" in successive SVD-based approximation has not been addressed in this version. (In the paper, the successive SVD approximation is performed in one direction, i.e., from 1 to S. But this is not the only direction possible.) I understand that investigating all the possible directions can be difficult, but I suggest adding this discussion with a few other example directions, which can motivate readers to try other possible combinations.


**Summary Of The Paper:**

This paper proposes a new tensor decomposition method for neural network compression. The main idea is to devise a decomposition structure that is composed of a sequence of Kronecker products, which generalizes most of the well-known tensor decompositions. Existing decompositions mostly rely on specific structures (of decomposition), which might not be best suited for network compression purposes. The method itself is quite simple (applying successive greedy binary Kronecker decompositions), which is a good point for the main goal. Experiments show that the proposed method achieves state-of-the-art performance for CIFAR-10, ImageNet, and super-resolution experiments.

**Summary Of The Review:**

I believe that the authors propose a flexible decomposition structure that is efficient for neural-net compression. SeKron might not add better characteristics with respect to the traditional tensor decomposition context, but it is surely more efficient for network compression. The previous criticism regarding the position of SeKron in the existing tensor decomposition literature is still not clearly resolved in this version, I suggest toning down and limiting the scope of discussion.

[After rebuttal] I'm satisfied with the answers and raise my score to 8. The authors have appropriately revised the paper, i.e., toned down regarding the generality of SeKron, focusing more on the paper's main focus, i.e., neural net compression.

---

> ### Author Response · Authors · 2022-11-18
> **Alternate directions for expansion and toning down on generality**
>
> A1:
>
> We appreciate your insight into the possibility of following other directions during the decomposition process. Please see the updated PDF file - we have added this Figure
>
> https://drive.google.com/file/d/1ONZxXM2ppW8dFWBX-GUAXlhgJsdjseZq/view?usp=sharing
>
> and discuss the  alternate directions in the appendix. (We refer to this as well in the body of the paper - second paragraph under section 3.2)
>
> A2:
>
> We have made some changes to the wording to try tone down on the "generality" of SeKron and rather emphasize the flexibility of the representation that ultimately leads to us finding more efficient representations Specifically, these changes have been made in these areas of the paper:
> 1. Abstract - we replace the "generalizes widely used methods..." statement with a statement that emphasizes that SeKron provides a flexible family of factorization structures that also covers the structures of TT, TR, CP, Tucker
> 2. Fourth paragraph of intro - we do similar re-wording as done to the abstract
> 3. Theorem 2 - We change the theorem statement by removing the word "generality" and make it more obvious that theorem states that factorization structures imposed by CP, TT, TR and Tucker can be achieved using SeKron.
>
>  We hope that we have satisfactorily addressed your comments regarding the toning down of the generality as well as incorporating the alternative directions for the decomposition. If this is the case we hope that you can increase your score to give us a chance to present our work at ICLR 2023.

---

> > ### Comment · Reviewer_Qck6 · 2022-11-20
> > **Thank you for the answers.**
> >
> > Thank you for the answers. I'm generally satisfied with the revision, but I have another small question. In Appendix B, the authors described that alternative directions are not covered by SeKron since it is difficult to apply the efficient convolution algorithm. However, in my understanding, the alternative directions can be similarly processed efficiently with a slight modification in the algorithm. Please correct me if I'm missing something.

---

> > > ### Author Response · Authors · 2022-11-21
> > > **Regarding alternate directions**
> > >
> > > Thank you for your comment. We have changed our wording in that section of the appendix to clarify this further (see: https://drive.google.com/file/d/1G2U--nPAH2OUlj9XkIFVwWUBiZZf1PK7/view?usp=sharing). Previously, we had the sentence "However, such alternative structures do not fall within our SeKron framework as they cannot make use of our convolution algorithm (Algorithm 2)". We agree that it is possible to modify Algorithm 2 to support alternate directions, but we wanted to make it clear that Algorithm 2 cannot be used as is. We hope that our updated explanation is more clear.

---

> > > > ### Comment · Reviewer_Qck6 · 2022-11-22
> > > > **Thank you for the update.**
> > > >
> > > > I'm satisfied with the authors' answers.

---

### Official Review · Reviewer_f4rp · 2022-11-02

**Confidence:** 3
**Correctness:** 4
**Technical Novelty And Significance:** 3
**Empirical Novelty And Significance:** 2
**Recommendation:** 6

**Clarity, Quality, Novelty And Reproducibility:**

**Clarity:** The paper is clearly written.

**Quality:** The derivations seem to be correct.

**Novelty:** Medium. The authors propose a new tensor decomposition for convolutional layers that generalizes many previous models.

**Reproducibility:** The authors do not provide code. The hyperparameters and details are not listed either.

**Strength And Weaknesses:**

**Strengths**

1. The authors propose a new tensor decomposition method based on SVD, which generalizes CP, Tucker and TT/TR.
2. The authors show how to parameterize kernels in CNN by the proposed SeKron format. The proposed method allows convolutional operations by operating each core tensors, without computing the full tensor. Therefore, the proposed method is efficient to do CNN inference.


**Weaknesses and Questions**

1. I do not catch how the authors do the compression. Do they use end-to-end training for the given SeKron format or just apply Algorithm 1 on pretrained CNNs? Is there any finetuning after the decomposition?
What devices do the authors use in experiments? Why do the authors only list CPU time? What about FLOPs and GPU time for SeKron and baselines?
2. The authors consider addressing the architecture limitations of tensor decomposition by proposing the flexible SeKron. However, there exists some work to tackle the problem by exploring many potential tensor networks. I think a good reference is [1], since it also focuses on the CNN compression by using genetic algorithms to search tensor network structures. Moreover, [2] proposes a permutation and rank search for TR format, which can be also applied if the CNN is not trained end-to-end.
3. As SeKron is developed based on the sum of Kronecker structure proposed by Hammed et al. (2022), can the authors provide more intuitions and advantages about using this sequence of sum of Kronecker in SeKron? Moreover, why did not the authors compare SeKron with the sum of Kronecker in experiments? I think it is an important baseline. Currently, the advantages of SeKron are not clear to me.
4. If I understand correctly, in Algorithm 2, the contraction or projection must be done sequentially for $i= S, \dots, 1$. Why do the authors say the projection can be performed independently? Moreover, will this sequential procedure increase computational time compared with traditional convolutional layers which are highly parallelized?

[1]. Hayashi, K., Yamaguchi, T., Sugawara, Y., & Maeda, S. I. (2019). Exploring unexplored tensor network decompositions for convolutional neural networks. Advances in Neural Information Processing Systems, 32.

[2]. Li, C., Zeng, J., Tao, Z., & Zhao, Q. (2022, June). Permutation Search of Tensor Network Structures via Local Sampling. In International Conference on Machine Learning (pp. 13106-13124). PMLR.


**Summary Of The Paper:**

This paper focuses on CNN compression using low-rank factorizations. In particular, the authors propose a new tensor decomposition called SeKron, which generalizes traditional TT, TR, CP and Tucker decomposition. The authors establish a SVD-based algorithm to decompose a given tensor into the SeKron format. Due to the special structure of SeKron, convolutional operations can be conducted without reconstructing the full tensor, which saves memory and FLOPs when doing inference in CNNs.

The authors conduct experiments on WideResNet/ResNet for classification and SRResNet/EDSR for super-resolution. They compare with several decomposition and pruning methods. The proposed model shows similar compression rates, but achieves better performance.

**Summary Of The Review:**

This paper proposes a new tensor decomposition method which can be used for CNN compression. The proposed method is flexible, which is potential for some future applications. However, the intuition and advantage is not clear to me currently. I would like to raise my score if the authors could address my concerns.

---

> ### Author Response · Authors · 2022-11-18
> **Addressing the training procedure and comparison with other baselines**
>
> **A1:**
>
> 1. We follow the same training procedures as the baselines we compare against, which involves performing the decomposition on weight tensors, followed by training for a certain number of epochs. We have added the following statement to the first paragraph of the Experimental results section to make this more clear: "In all experiments we
> compress convolution layers of pre-trained networks using various compression approaches and then
> re-train the resulting compressed models. We provide implementation details in Appendix"
> 2. We have added a description of our implementation details including which devices were used for training to the Appendix. Thank you for pointing this out.
> 3. As stated in the introduction, our motivation is to compresses models for use on edge devices, which typically use CPUs to perform computations and not GPUs, that is why we report CPU latency. Moreover, we did not report FLOPs as they are only a proxy for the latency, which we report directly via CPU time. (See answer to q4 as well discussing GPU vs CPU)
>
> **A2:**
>
> The works of [1] and [2] are mainly focused on a searching through a space of tensor networks to maximize network accuracy (former) or minimize tensor approximation errors (latter).
>
> **Comparison with [1]:**
>
> Our work introduces a particular family of tensor networks that share two interesting properties:
> 1. All SeKron structures share the same decomposition algorithm (Algorithm 1) meaning we can recover values of the decomposed tensor exactly.
> 2. All SeKron structures share a single convolution algorithm (Algorithm 2) that can be carried out using efficient built-in operations such as Conv3D.
>
> The first distinguishing factor between our work and [1] is that no such decomposition algorithm is presented for the enumerated tensors of [1]. Moreover, the SeKron family is designed to leverage existing efficient operations such as Conv3D, whereas the contractions of tensors in [1] do not leverage such operations.
>
> The second distinction between our work and theirs relates to their search criteria. The search in [1] is optimizing for **accuracy** whereas we are optimizing for **latency**. Although genetic algorithms can be used in our approach as well, we were able to search the entire space and select a fast structure that best satisfies our desired compression rate (Fig.3). This is because measuring latency can be done quite quickly as opposed to measuring network accuracy in [1], which requires training a model.
>
> **Comparison with [2]:**
>
> The main distinction between our work and [2] is that we introduce a family of factorization structures (SeKron) whereas [2] introduces a procedure to search for optimal permutations of a given factorization.
>
> Nonetheless, we appreciate you highlighting these works and will refer to them in the Related Work section.
>
> **A3:**
>
> In the paper we highlight two main reasons (theoretical and experimental) as to why using a sequence is beneficial:
>
> 1. In Figure (1b) we show that there exists tensors that can be represented using fewer parameters when using a larger sequence length.
> 2.  More importantly, in Figure 3 we show experimentally that there is a severe limitation when not using sequences. That is, there exists only 3 viable compression rates when decomposing a tensor using the method of Hameed et al. (2022) vs. 129 viable compression rates when simply opening up the search space to a sequence length of 3. We explain in the paper that the "viable" region is the region consisting of factorization structures that do not increase the CPU latency during the forward pass in comparison to the baseline.
>
> Finally, we do in fact compare to sum of Kronecker. See "Binary Kronecker" Hameed et al. (2022) in Table 2.
>
>
> **A4:**
>
> Your understanding about the computation needing to be performed sequentially is correct. By "independently" we are referring to the ability to separate the convolution into multiple sequential convolutions - in contrast to having to reconstruct the approximate tensor (implicitly represented by the factors) in order to perform the convolution operation. The sequential procedure may increase run-time on devices favouring parallel computations such as GPUs indeed. However, as we state in the introduction, our motivation is to find factorization structures that are effective on edge-devices which typically use sequential computations on CPUs. Therefore we report CPU latency in our experiments.
>
>
> We hope that we have addressed your comments regarding the training procedure and comparison with other baselines adequately. If this is the case we hope that you can increase your score to give us a chance to present our work at ICLR 2023.

---

> > ### Comment · Reviewer_f4rp · 2022-11-21
> > **Response to the answer**
> >
> > Thanks for the authors’ answer and adding more details. I think the response addresses my concerns. I will update the score to 6.

---

### Decision · Program_Chairs · 2023-01-20

**Decision:**

Reject

**Justification For Why Not Higher Score:**

Theorem 1 and 2 do not provide clear and significant findings, which can not support the proposed method well.  The paper doesn't sufficiently discuss the limitations of the Kronecker product structure. The clarity of paper could be further improved.

**Justification For Why Not Lower Score:**

N/A

**Metareview: Summary, Strengths And Weaknesses:**

Summary: This paper proposes a tensor decomposition method for neural network compression. The main idea is to devise a decomposition structure that is composed of a sequence of Kronecker products, which generalizes most of the well-known tensor decompositions.  Experiments show that the proposed method achieves state-of-the-art performance for CIFAR-10, ImageNet, and super-resolution experiments.

Strengths: The authors propose a new tensor decomposition method based on SVD, which generalizes CP, Tucker and TT/TR. The proposed method allows convolutional operations by operating each core tensors, without computing the full tensor. Therefore, the proposed method is efficient to do CNN inference. It is definitely more flexible in determining the shapes of the factors. For network compression purposes, this can be a better choice.

Weakness: SeKron is claimed to generalize various other decompositions. But it is not clear that the proposed algorithm could ever reproduce those decompositions. The clarity of the paper needs to further improved regarding the notations.  The novelty of the proposed method is relatively weak since the sum of Kronecker structure proposed by Hammed et al. (2022) and tensor decomposition for NN compression is widely studied. It lacks strong motivation and deep insight why SeKron has significant impacts on NN compression.